# NaDRO: Leveraging Dual-Reward Strategies for LLMs Training on Noisy Data

Haolong Qian[1,*] Xianliang Yang[2], Ling Zhang[2], Lei Song[2], Jiang Bian[2], Chun Yuan[1,†]

[1]Tsinghua Shenzhen International Graduate School, Tsinghua University
[2]Microsoft Research Asia, Microsoft
qhl23@mails.tsinghua.edu.cn

## Abstract

Group Relative Policy Optimization (GRPO) fine-tuning has demonstrated significant enhancements in reasoning tasks. However, it often relies on high quality labeled dataset, which is typically difficult to obtain. To address this challenge, we introduce **N**oise-**A**ware **D**ual-**R**eward **O**ptimization (**NaDRO**) to effectively enhances the training of Large Language Models (LLMs) under noisy or ambiguous supervision. NaDRO operates through two key components: **(1) Preference-based Outcome Reward (POR)**, which makes a principled bias-variance tradeoff, reducing training variance by learning from robust preference rankings instead of overfitting to single-best estimates; and **(2) Context Perception Reward (CPR) mechanism**, which ensures that LLMs conduct necessary qualitative assessment of the current problem state to foster deeper situational understanding prior to decision-making. To validate our approach in a realistic decision-making testbed, we model classic combinatorial optimization problems like the Traveling Salesman Problem (TSP) and Capacitated Vehicle Routing Problem (CVRP) as Markov Decision Processes, generating training data via cost-limited exploration. Our results demonstrate that the fine-tuned Qwen 7B and Llama 3.1-8B models achieve statistically robust performance, significantly outperforming leading LLM baselines and standard fine-tuning methods on these complex benchmarks. Code is released at https://github.com/microsoft/HeurAgenix/tree/NaDRO.

$$R_t = w_{CPR} \cdot R_{CPR}(s_t, a_t) + w_{QRP} \cdot R_{QRP}(s_t) + w_{other} \cdot R_{other}(s_t)$$

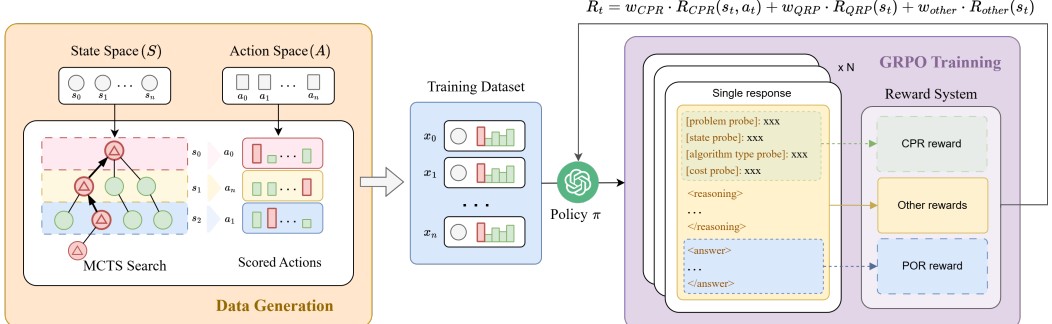

Figure 1: Training LLMs with NaDRO: Leveraging Noisy MCTS Data through Preference-based Outcome Reward (POR) (see Section 4.1) and Context Perception Reward (CPR) (see Section 4.2)

---

*The work was conducted during Haolong's internship at Microsoft Research Asia.
†Corresponding author.

39th Conference on Neural Information Processing Systems (NeurIPS 2025).

# 1 Introduction

Large Language Models (LLMs) exhibit considerable promise in complex reasoning [1–4] but encounter notable challenges in long-horizon decision making tasks [5, 6]. This challenge stems from a fundamental ambiguity inherent [7] in such tasks: often, there is no single optimal action, and even if one exists, its true long-term value is nearly impossible to perfectly evaluate with finite computational resources. This reality creates a critical mismatch for existing fine-tuning techniques like Group Relative Policy Optimization (GRPO) [8]. Applying such methods directly to these ambiguous, imperfect reward signals leads to training instability and suboptimal performance [9].

To address this, we propose Noise-Aware Dual-Reward Optimization (NaDRO), a method enabling LLMs to learn effectively from noisy data [10], particularly within offline training paradigms such as GRPO [8]. NaDRO is founded upon two core mechanisms:

- **Preference-based Outcome Reward (POR):** This component makes a principled bias-variance tradeoff. Instead of overfitting to a single, high-variance 'optimal action', it reduces training variance by learning from the more stable preference rankings of multiple candidate actions [11, 12, 10].
- **Context Perception Reward (CPR):** As a process reward, CPR incentivizes LLMs to accurately assess key qualitative features of the current problem state prior to reasoning and action selection. This provides a dense and immediate training signal [13, 14], counteracting the sparse, delayed rewards common in long-horizon tasks and mitigating the risk of "reward hacking".

To evaluate NaDRO's ability to handle noisy data in complex, long-horizon tasks, we model classic combinatorial optimization problems, such as the Traveling Salesman Problem (TSP) and the Capacitated Vehicle Routing Problem (CVRP), as Markov Decision Processes. In this setup, the LLM learns to act as a meta-controller, dynamically selecting heuristic algorithms. By generating training data via Monte Carlo Tree Search (MCTS), we effectively simulate the noisy and imperfect data conditions prevalent in specialized real-world domains. Our extensive experiments robustly show that small-sized models like Qwen 7B [15] and Llama 3.1-8B [16], when fine-tuned with NaDRO, significantly surpass the performance of leading LLMs, including GPT-4o and DeepSeek R1, on these demanding decision-making tasks.

Our primary contributions are:

1. **A Novel Paradigm for Noise-Robust LLM Training:** NaDRO offers a new methodology for LLMs to effectively utilize large-scale, noisy exploration data in complex decision-making task, reducing reliance on perfectly labeled data.
2. **Synergistic Noise Reduction via Dual Rewards:** NaDRO, through POR and CPR, synergistically optimizes learning signals from both outcome preference and intermediate reasoning process, enhancing learning efficiency and decision reliability in noisy environments.
3. **Superior Empirical Performance:** We demonstrate that moderately-sized LLMs like Qwen 7B and Llama 3.1-8B, when trained with NaDRO, can outperform leading LLMs on challenging TSP and CVRP tasks.
4. **Broad Generalization Potential:** This research offers significant insights for applying and training LLMs in other complex decision-making domains characterized by difficult data acquisition, high labeling costs, or inherent data noise.

# 2 Related Work

## 2.1 LLM Fine-Tuning for Reasoning and Decision-Making

Fine-tuning strategies are crucial for adapting LLMs to specialized reasoning tasks. Methods such as Reinforcement Learning from Human Feedback (RLHF) [12, 11, 17] and Direct Preference Optimization (DPO) [18] have advanced model alignment. Concurrently, Group Relative Policy Optimization (GRPO) [8] leverages comparisons among multiple responses for logical reasoning tasks. This paradigm is increasingly applied to learning heuristic meta-policies [5] for sequential decision-making [19], where LLMs now act as meta-controllers. A persistent challenge across these

approaches is their reliance on high-quality reward signals, as performance degrades on the noisy data common in long-horizon tasks. Parallel to fine-tuning, alternative post-training approaches [20–22] also aim to enhance LLM capabilities.

## 2.2 Test-Time Scaling Method

Complementary to training-time improvements, Test-Time Scaling (TTS) [23] is an inference-time strategy that enhances performance by exploring multiple candidate solutions [24, 25]. While NaDRO focuses on improving the underlying policy quality during training, TTS can be applied during inference to further refine the final output by leveraging additional computation.

# 3 Preliminaries

**MDP Formulation** We model the task of dynamic heuristic selection as a Markov Decision Process (MDP). Within this framework, an LLM acts as a meta-controller policy, $\pi(a_t|s_t)$, that maps the current problem state $s_t$ to a heuristic action $a_t$ from a predefined discrete set $A$. Applying the action transitions the environment to a new state $s_{t+1}$. The objective is to learn a policy that optimizes the quality of the final solution after a sequence of $H$ decisions.

**Data Generation and Inherent Noise** To train the LLM policy, we require state-action value pairs as supervisory signals. We employ Monte Carlo Tree Search (MCTS) to explore the decision space from any given state $s_t$ and generate an estimated value, $Q(s_t, a_i)$, for each candidate action $a_i$. However, due to computational constraints and the credit assignment problem inherent in long-horizon tasks, these $Q$-values are fundamentally noisy and imperfect approximations of an action's true long-term utility. A critical observation, empirically validated in Figure 2, is that while the absolute scores from MCTS are unreliable, their relative rankings preserve a much more robust signal. The results show that a policy constrained to select actions from the top 30% of MCTS evaluated scores significantly outperforms a random policy and achieves near optimal performance. This finding strongly suggests that an effective learning paradigm should focus on leveraging these robust preference signals rather than attempting to directly regress the noisy, absolute action values.

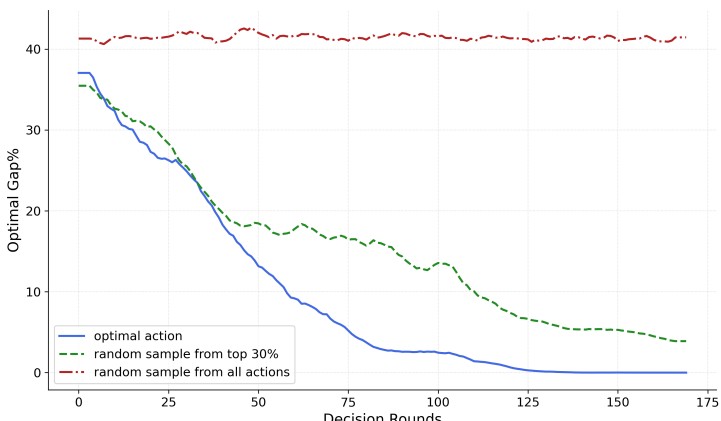

Figure 2: Preference learning effectiveness on noisy data in rd100 task. The y-axis represents the optimality gap of the final solution. The optimal action policy always selects the action with the highest MCTS score, while the other policies sample from either the top 30% of actions or all actions at each step.

# 4 Methodology

The GRPO algorithm, which is a Reinforcement Learning with Verifiable Rewards (RLVR) paradigm, is fundamentally challenged when applied to complex decision-making tasks. Unlike domains with clear, ground-truth answers such as mathematics, these tasks often lack a single correct action,

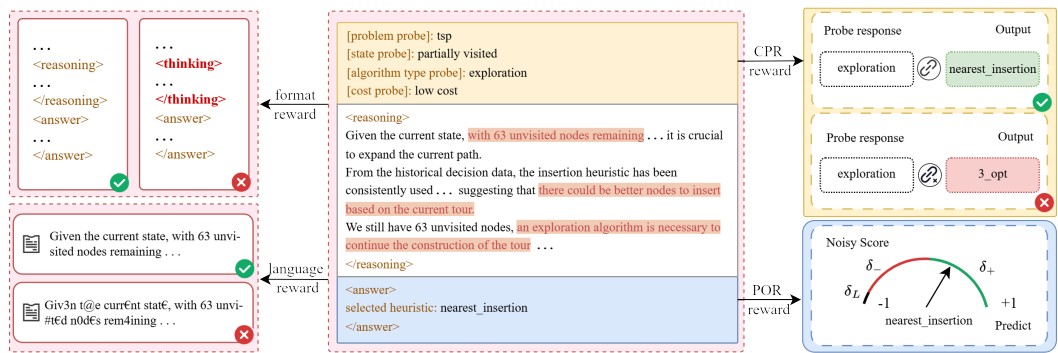

Figure 3: Schematic of Reward Components for LLM Training with NaDRO, detailing Format Reward [8], Language Consistency Reward [26], and the operational mechanisms of the novel POR and CPR.

and any feasible evaluation of action quality is inherently noisy. Directly applying the standard GRPO approach to reward the highest value is therefore prone to training instability. To address this mismatch, we introduce NaDRO framework to enhance the learning process through its two core contributions: redesigning the outcome reward to learn from robust preference rankings (POR) and supplementing it with a process reward for ambiguous outcomes (CPR). This dual-reward mechanism is complemented by standard auxiliary signals, such as format rewards [8] and language consistency rewards [26].

## 4.1 Preference-based Outcome Reward (POR): Learning from Ordered Noisy Signals

To stabilize learning from noisy MCTS values, we introduce the POR method to partition the ranked action space using a hyperparameter $k \in (0, 1)$ into a positive set $A_+$ (top $k\%$) and a negative set $A_-$. An action's reward is then linearly scaled based on its rank $i$ within action space $A$:

$$R_{\text{POR}}(s_t, a_t) = \begin{cases} +R_p \cdot \left(1 - \frac{i-1}{\lfloor k \cdot N_A \rfloor}\right), & \text{if } a_t \in A_+ \\ -R_n \cdot \frac{i - \lfloor k \cdot N_A \rfloor}{N_A - \lfloor k \cdot N_A \rfloor}, & \text{if } a_t \in A_- \end{cases} \tag{1}$$

This design is grounded in the bias-variance tradeoff. A standard Top-1 approach uses a high-variance learning target (a sharp Dirac delta function, $\delta(a = a^*)$) that, while unbiased in expectation, jumps erratically between batches and destabilizes training. POR replaces this with a softer target over the positive set, $U(a|A_+)$, which significantly reduces training variance. This is achieved by introducing a minor bias, as the positive set $A_+$ serves as a robust proxy for the true optimal actions $a^*$. This principled tradeoff prioritizes learning stability over unbiasedness, a necessary adaptation when dealing with inherently ambiguous or noisy evaluations.

Furthermore, the selection of the hyperparameter $k$ is also itself principled. We derive its optimal value by maximizing a value separability term, defined as the expected value gap between the positive and negative action sets. We formalize this objective as:

$$J(k) = \mathbb{E}\left[\frac{1}{N_{A_-}(k)} \sum_{a \in A_-(k)} Q(a) - \frac{1}{N_{A_+}(k)} \sum_{a \in A_+(k)} Q(a)\right] \tag{2}$$

By modeling the relative action values based on MCTS, we can compute an estimate for $J(k)$ for any given partition. The theoretically optimal partition is therefore found by solving for $k^* = \arg\max_k J(k)$. As demonstrated in Section 5.3, this theoretical optimum aligns remarkably well with our empirical findings, reinforcing the principled design of our framework.

## 4.2 Context Perception Reward (CPR): Unearthing Richer Process Rewards

Dependence exclusively on outcome rewards encounters significant challenges. Especially in complex long-horizon decision tasks, a valid reasoning process or accurate qualitative perception of the problem's present state typically underpins effective decision-making. To address this, we introduce the CPR mechanism, designed to guide and evaluate the LLMs' pre-decision cognitive process and process-level rewards. Crucially, CPR provides a dense, immediate, and outcome independent learning signal, which helps mitigate the severe credit assignment problem and prevents the model from "reward hacking" by focusing on the reasoning process.

Specifically, the CPR mechanism requires the model, prior to its standard reasoning, to first conduct qualitative evaluations of key contextual dimensions, including problem type, problem state, current cost and other relevant features. The model must clearly output its assessment of these dimensions.

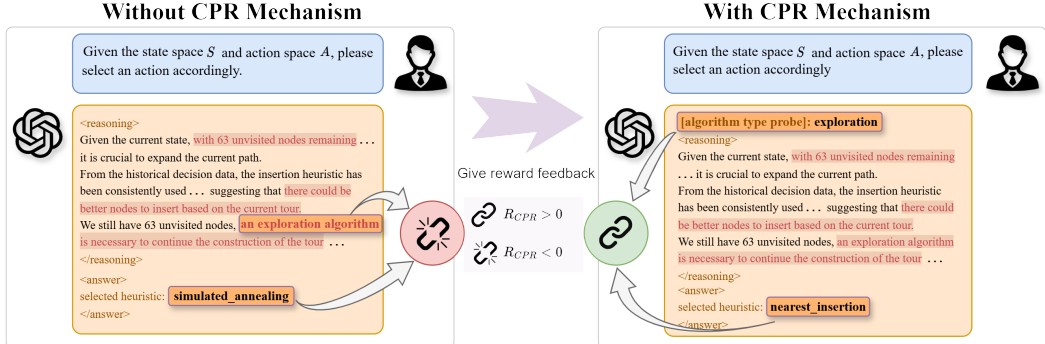

Figure 4: Impact of the CPR mechanism in rectifying qualitative judgment errors during LLM inference. The illustration shows how CPR guides the LLM towards accurate contextual assessments, mitigating errors that might otherwise go uncorrected.

As illustrated in Figure 4, even if the final decision happens to be correct or good, the model should not receive the full reward if it failed to establish a correct understanding of the state. To quantify the quality of the cognitive process, we define a CPR reward function for each state $s_t$:

$$R_{\text{CPR}}(s_t) = \sum_{j=1}^{n} \left[ \mathbb{I}\big(\hat{y}_{t,j} = y_{t,j}^*\big)\, r_j^+ \; - \; \mathbb{I}\big(\hat{y}_{t,j} \neq y_{t,j}^*\big)\, r_j^- \right] \tag{3}$$

Here, $j$ corresponds to the context perception dimensions, $\hat{y}_{t,j}$ denotes the model's response for the $j$-th contextual dimension at state $s_t$, and $y_{t,j}^*$ is the ground-truth label, which can be easily annotated due to its qualitative nature. We employed a rule-based method for annotation, details can be found in Appendix B. Reward weights assigned to correct and incorrect answers are denoted by $r_j^+$ and $r_j^-$, respectively.

Through the CPR mechanism, the LLMs are incentivized to learn how to accurately perceive and interpret the current state features of the problem. $R_{\text{CPR}}$ provides a learning signal that is independent of the final outcome, more immediate, and more stable.

## 5 Experiments

The NaDRO framework is developed to enhance LLM fine-tuning for a wide array of long-horizon decision-making tasks. To empirically evaluate its effectiveness and demonstrate its capabilities, this study primarily focuses on heuristic algorithm selection in the context of TSP. Furthermore, to illustrate the generalizability of our method, we also present experiments conducted on CVRP, details can be found in Appendix A. The core objectives of these evaluations are to rigorously assess NaDRO's ability to leverage MCTS data for robust policy learning and to benchmark its performance.

## 5.1 Experimental Settings

### 5.1.1 Tasks and Datasets

The primary task for LLMs is to act as a meta-controller, dynamically selecting the most appropriate heuristic algorithm at each decision step to progressively construct or improve solutions for TSP and CVRP instances.

- **TSP:** We randomly selected a diverse set of 10 instances from the well-known TSPLIB benchmark library to evaluate our method. These instances were chosen to cover a range of sizes and complexities, ensuring that the evaluation captures both scalability and robustness to diverse structural patterns.

- **CVRP:** We selected instances 1 through 10 from the Golden dataset for CVRP. These instances are particularly challenging, characterized by a large number of customers ($n$) ranging from 200 to 480 and vehicle fleet sizes ($K$) varying between 5 and 16. Detailed results can be found in Appendix A.

For each instance, a decision sequence consists of $H$ steps, equal to $N/n_{run}$ where $N$ is number of nodes, $n_{run}$ is the running times of the heuristics selected at each step. At each step $t$, LLMs observe the current state $s_t$ and selects an action $a_t$ from a predefined action space (the full state space and action space are described in Appendix E and F).

### 5.1.2 Models and Baselines

We employed the **Qwen2.5-7B-Instruct-1M** [15] and **Llama-3.1-8B-Instruct** [16] as base LLMs for fine-tuning with our NaDRO method. To demonstrate its efficacy, we compared the performance of NaDRO-Qwen 7B and NaDRO-Llama 3.1-8B against several baselines, including GPT-4o [27], GPT-o3, Deepseek-R1 [26]; Mainstream heuristic and meta-heuristic methods like OR-Tools, LKH [28], GLS [29], ACO [30], EoH [31], ReEvo [32] are also compared.

### 5.1.3 Training Data Generation

Training data in the form of $(s_t, \{a_i, Q(s_t, a_i)\}_{i=1}^{N_A})$ pairs were generated using MCTS. For each state encountered during a training instance generation process, MCTS was run for 1000 iterations to evaluate the set of $N_A$ candidate heuristic actions. The default policy for MCTS rollouts was random heuristic selection. This process inherently produces noisy $Q$ values due to the limited search budget, as discussed in Section 3.

### 5.1.4 Fine-tuning and Implementation Details

All models were fine-tuned using the GRPO algorithm. For NaDRO-Qwen 7B, NaDRO-Llama 3.1-8B and the GRPO baseline, fine-tuning was performed with number of generations set to 12. Details of the fine-tuning experiments are provided in Appendix D. Experiments were performed on a cluster with NVIDIA A100 and NVIDIA A6000 GPUs, leveraging the Unsloth framework [33] for optimized training efficiency. A typical fine-tuning run for our primary models took approximately 32 hours. Comprehensive fine-tuning procedures are further detailed in Appendices C and D.

## 5.2 Main Empirical Results on TSP

To evaluate the efficacy of our proposed NaDRO method, we conducted extensive experiments on Traveling Salesman Problem. The comparative results, presented in Table 1, demonstrate the clear superiority and statistical robustness of the NaDRO framework.

Table 1: Comparative optimality gap (%) on TSP instances, sorted by problem size. Results for NaDRO-Qwen 7B and NaDRO-Llama 3.1-8B are generated using our proposed NaDRO mechanism. For baselines, the performance of ReEvo [32] and EoH [31] is based on their original code implementations.

| Method | pr76 | gr96 | rd100 | ch130 | pr152 | u159 | brg180 | gr202 | tsp225 | a280 |
|---|---|---|---|---|---|---|---|---|---|---|
| *Traditional Solvers* | | | | | | | | | | |
| LKH | 0.00 ± 0.0 | 0.00 ± 0.0 | 0.00 ± 0.0 | 0.00 ± 0.0 | 0.00 ± 0.0 | 0.00 ± 0.0 | 0.00 ± 0.0 | 0.00 ± 0.0 | 0.01 ± 0.0 | 0.01 ± 0.0 |
| OR-Tools | 2.58 ± 0.0 | 3.11 ± 0.0 | 3.93 ± 0.0 | 0.51 ± 0.0 | 2.92 ± 0.0 | 3.14 ± 0.0 | 0.51 ± 0.0 | 5.12 ± 0.0 | 5.13 ± 0.0 | 6.32 ± 0.0 |
| *Metaheuristics* | | | | | | | | | | |
| GLS | 1.05 ± 0.4 | 1.65 ± 0.8 | 7.48 ± 2.3 | 5.32 ± 2.1 | 2.82 ± 0.9 | 4.22 ± 1.4 | 3.58 ± 2.5 | 5.44 ± 0.3 | 3.74 ± 1.2 | 2.83 ± 1.7 |
| EoH + GLS | 1.31 ± 0.3 | 1.34 ± 0.4 | 3.85 ± 1.2 | 5.64 ± 2.4 | 2.73 ± 0.7 | 4.07 ± 0.6 | 4.35 ± 2.3 | 3.31 ± 1.1 | 2.56 ± 0.1 | 4.04 ± 0.6 |
| ACO | 9.26 ± 1.1 | 12.58 ± 2.6 | 17.12 ± 2.9 | 8.89 ± 1.4 | 4.70 ± 1.6 | 9.28 ± 1.5 | 3.43 ± 2.4 | 10.43 ± 2.7 | 9.71 ± 2.1 | 17.78 ± 1.3 |
| ReEvo + GLS | 1.86 ± 0.6 | 2.37 ± 1.2 | 6.70 ± 2.1 | 5.04 ± 1.9 | 2.88 ± 1.6 | 3.99 ± 1.5 | 4.11 ± 2.3 | 5.80 ± 0.7 | 4.73 ± 1.6 | 3.92 ± 0.5 |
| ReEvo + ACO | 7.71 ± 0.1 | 6.66 ± 1.4 | 9.12 ± 0.9 | 5.85 ± 1.4 | 7.87 ± 0.5 | 5.52 ± 0.1 | 3.91 ± 2.3 | 8.29 ± 1.4 | 6.99 ± 2.9 | 15.91 ± 2.1 |
| *LLM-based Approaches* | | | | | | | | | | |
| GPT-4o | 0.99 ± 0.7 | 1.55 ± 0.3 | 2.18 ± 0.7 | 3.19 ± 0.3 | 1.29 ± 0.6 | 2.48 ± 0.7 | 4.70 ± 0.6 | 4.23 ± 1.2 | 5.78 ± 0.6 | **7.11 ± 1.1** |
| GPT-o3 | 0.37 ± 0.3 | 0.53 ± 0.4 | 0.44 ± 0.5 | 2.86 ± 0.2 | 0.28 ± 0.2 | **0.68 ± 0.3** | 2.67 ± 0.3 | 3.79 ± 0.5 | 5.11 ± 0.6 | **3.71 ± 0.4** |
| Deepseek-R1 | 0.69 ± 0.3 | 1.04 ± 0.2 | 1.18 ± 0.4 | 2.27 ± 0.8 | 0.86 ± 0.2 | 1.83 ± 0.5 | 2.94 ± 0.5 | 3.88 ± 0.5 | 6.47 ± 0.9 | 14.10 ± 0.7 |
| Qwen 7B | 2.33 ± 0.4 | 3.89 ± 0.4 | 3.98 ± 1.1 | 4.45 ± 0.3 | 1.31 ± 0.2 | 3.29 ± 0.5 | 5.86 ± 1.9 | 15.38 ± 3.3 | 10.42 ± 1.9 | 8.83 ± 1.4 |
| GRPO on Qwen 7B | 1.23 ± 0.2 | 1.97 ± 0.2 | 2.43 ± 0.2 | 2.29 ± 0.3 | 1.04 ± 0.1 | 2.57 ± 0.4 | 5.42 ± 0.3 | 14.68 ± 0.4 | 11.14 ± 0.3 | 7.33 ± 0.4 |
| Llama 8B | 1.79 ± 0.2 | 4.47 ± 0.8 | 3.84 ± 0.5 | 4.86 ± 0.5 | 2.07 ± 0.4 | 3.51 ± 0.4 | 7.39 ± 0.8 | 13.43 ± 1.1 | 9.92 ± 1.3 | 9.21 ± 1.2 |
| GRPO on Llama 8B | 0.67 ± 0.1 | 2.39 ± 0.5 | 2.12 ± 0.8 | 3.91 ± 0.7 | 0.89 ± 0.2 | 2.08 ± 0.5 | 5.22 ± 0.5 | 11.33 ± 0.7 | 7.33 ± 1.1 | 8.10 ± 0.6 |
| **NaDRO-Llama(Ours)** | **0.31 ± 0.1** | 0.28 ± 0.1 | 0.70 ± 0.2 | 2.19 ± 0.3 | 0.37 ± 0.1 | 0.94 ± 0.2 | 2.17 ± 0.3 | 2.41 ± 0.5 | 4.89 ± 0.8 | 6.22 ± 0.4 |
| **NaDRO-Qwen(Ours)** | 0.63 ± 0.2 | **0.16 ± 0.2** | **0.11 ± 0.1** | **1.80 ± 0.2** | **0.22 ± 0.1** | 1.38 ± 0.3 | **0.08 ± 0.1** | **2.29 ± 0.3** | **3.76 ± 0.5** | 5.31 ± 0.7 |

The results in Table 1 demonstrate the clear superiority of our NaDRO framework. Our NaDRO-tuned models consistently and significantly outperform other LLM-based approaches, including larger models like GPT-4o, and are highly competitive with established metaheuristics. The most critical insight comes from the direct comparison against standard GRPO fine-tuning on identical base models (Qwen 7B and Llama 3.1-8B). This comparison reveals decisive performance gains (for instance on the gr202, the optimality gap is reduced from 14.68% with standard GRPO to just 2.29% with NaDRO) that are directly attributable to our novel reward architecture, not merely the act of fine-tuning. Furthermore, the low standard deviations across multiple runs confirm that these improvements are statistically robust. While a specialized solver like LKH defines the performance ceiling, NaDRO's ability to guide a general-purpose LLM to surpass many strong metaheuristics highlights its practical efficacy for tackling complex combinatorial optimization problems.

## 5.3 Correctness Reward Dynamics and Robustness

To validate the robustness and practical applicability of the NaDRO framework, we analyzed the training dynamics of the answer correctness reward across different configurations. This analysis dissects the framework's sensitivity to the POR cutoff parameter k and its resilience under extreme data corruption.

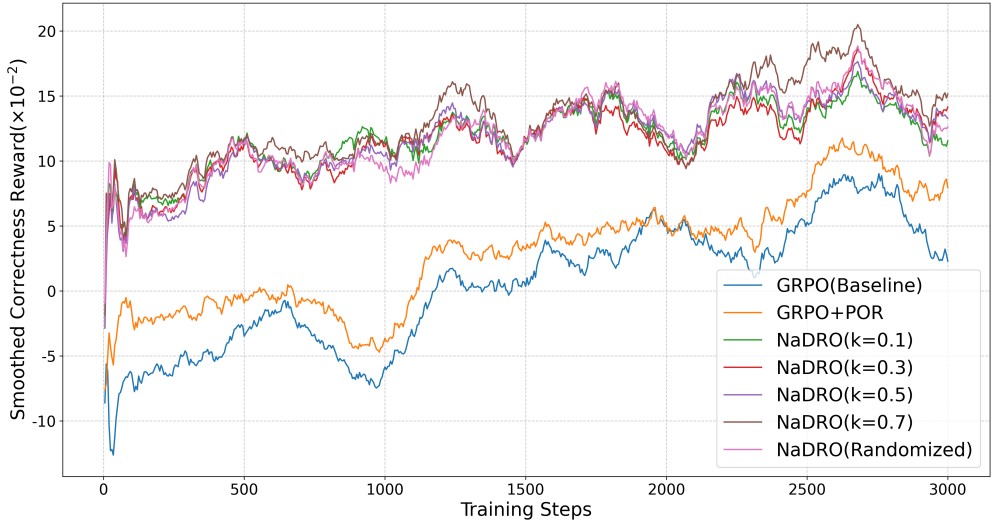

Figure 5: Correctness reward dynamics during training on the pr152 instance. The y-axis represents the smoothed correctness reward for the CPR mechanism which is scaled by a factor of $10^{-2}$.

The training dynamics presented in Figure 5 offer critical insights into the NaDRO framework. Compared to the GRPO baseline, the introduction of POR (GRPO+POR) yields a modest improvement; however, being fundamentally outcome-oriented, its learning dynamic remains limited. The introduction of the CPR mechanism yields a significant performance improvement, as seen in all NaDRO configurations, which rapidly achieve and maintain a high level of correctness, forming a distinct and vastly superior performance cluster.

Additionally, within the top cluster, the performance curves for a wide range of POR cutoff values (k from 0.1 to 0.7) are tightly grouped, confirming that NaDRO is not sensitive to this hyperparameter. And the framework's resilience is powerfully demonstrated by the NaDRO(Randomized) curve, which simulates a 10% chance of catastrophic data corruption at each step. Even under this severe noise condition, the model's ability remains almost unimpaired. This underscores NaDRO's exceptional resilience and its suitability for training on the highly imperfect data found in real-world scenarios.

The empirical observation that performance peaks around $k = 0.7$ is not coincidental but is a direct consequence of the task's inherent value structure, which we can analyze theoretically. This data-driven insight allows us to make a principled bias-variance tradeoff. As discussed in Section 4.1, choosing a larger $k$ intentionally introduces a bias towards a broader set of positive actions. This specific bias, guided by the training data, stabilizes the learning process by reducing variance.

To formalize this, we analyze the value distribution across action ranks. For our TSP task ($N_A = 15$), we define the value decrease for a given state $s$ as the difference between the action value at rank $i + 1$ and rank $i$: $\Delta C_i(s) = \mathbf{E}_s[Q_{i+1}(s) - Q_i(s)]$. The Contribution Ratio for each rank transition, visualized in Figure 6, is then the expectation of the softmax-normalized cost increases over the entire dataset $\mathcal{D}$:

$$\bar{p}_i = \mathbb{E}_{s \sim \mathcal{D}} \left[ \frac{\exp(\Delta C_i(s))}{\sum_{j=1}^{N_A - 1} \exp(\Delta C_j(s))} \right] \quad (4)$$

This value, $\bar{p}_i$, represents the average relative magnitude of the performance drop occurring between rank $i$ and $i + 1$, normalized across all transitions. Figure 6 visualizes these average ratios.

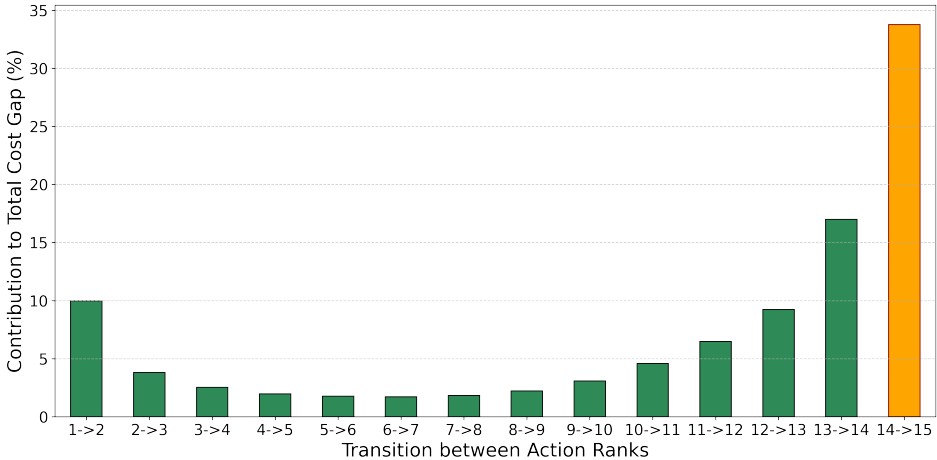

Figure 6: Average Contribution Ratios Across Rank Transitions (TSP Task). Each bar represents the average percentage of the total value gap attributable to the drop between adjacent ranks.

Figure 6 reveals a highly non-uniform value distribution specific to our TSP task dataset: the performance drop-off between the worst-ranked actions is far more significant than the differences among the top-ranked ones. For instance, the transition from rank 14 to 15 alone accounts for over 33% of the total gap. This particular distribution suggests that, for this task, prioritizing the avoidance of the worst actions is critical. By applying this relative value model to our Value Separability function $J(k)$, we can calculate the expected separability for every possible cutoff, as shown in Table 2.

Table 2: Optimal Cutoff Determination via Value Separability Analysis

| k | 1 | 2 | 3 | 4 | 5 | 6 | 7 | 8 | 9 | 10 | 11 | 12 | 13 | 14 |
|---|---|---|---|---|---|---|---|---|---|----|----|----|----|----|
| **k/N$_A$ (%)** | 6.7 | 13.3 | 20.0 | 26.7 | 33.3 | 40.0 | 46.7 | 53.3 | 60.0 | 66.7 | **73.3** | 80.0 | 86.7 | 93.3 |
| **J(k)** | 0.3486 | 0.3809 | 0.3957 | 0.4071 | 0.4173 | 0.4280 | 0.4398 | 0.4550 | 0.4847 | 0.5106 | **0.5284** | 0.5262 | 0.4901 | 0.3953 |

The utility function $J(k)$ reaches its maximum at $k = 11$, corresponding to an optimal positive set ratio of $11/15 \approx 73.3\%$. This theoretical result provides a strong justification for our empirical finding that performance peaks around $k = 0.7$.

## 5.4 Training Stability: Loss and Variance Analysis

Beyond final task performance, a critical indicator of a robust training framework is the stability of its optimization process. To provide empirical support for the theoretical motivations presented in Section 4, we analyzed the training dynamics of our framework, focusing on two key metrics: reward variance and training loss. These analyses reveal how NaDRO's dual-reward mechanism fosters a more stable and effective learning process compared to baseline methods.

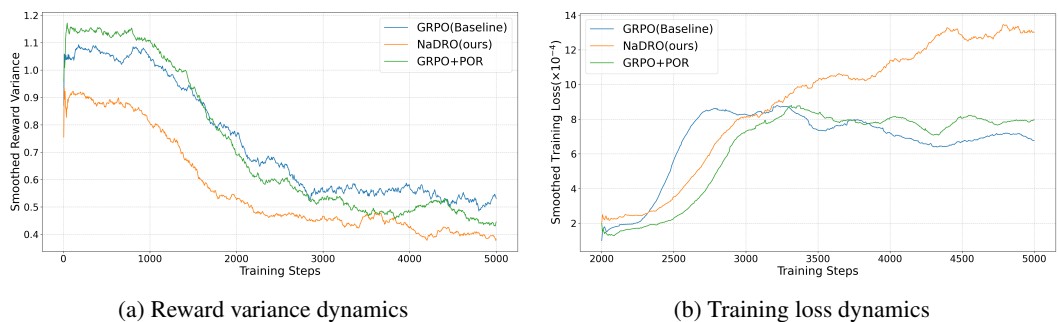

(a) Reward variance dynamics          (b) Training loss dynamics

Figure 7: Training stability analysis. (a) The variance of the total reward signal over training steps. The NaDRO framework demonstrates significantly lower and more stable reward variance compared to the volatile baseline GRPO. (b) Training loss dynamics. The baseline GRPO's loss rapidly collapses to a near-zero value, indicative of "reward hacking," while NaDRO maintains a healthier, actively fluctuating loss. The y-axis values are scaled by a factor of $10^{-4}$.

Our analysis of these dynamics, presented in Figure 7. First, Figure 7(a) shows that the NaDRO framework significantly reduces reward variance compared to the standard GRPO baseline. This result provides direct empirical validation for the core theoretical principle behind POR (discussed in Section 4.1), by rewarding a preference subset our method generates a more stable learning signal, which is fundamental for effective learning in noisy environments.

Furthermore, the training loss curves in Figure 7(b) offer a critical insight into the quality of the learning process. The standard GRPO model's loss quickly drops to a near-zero value and stagnates, a classic symptom of "reward hacking" where the model exploits the reward function without learning a robust strategy. In stark contrast, the NaDRO model maintains a healthier, actively fluctuating loss. This dynamic, paired with its high final reward, proves that the model remains meaningfully engaged in learning rather than collapsing into a trivial solution. Together, these findings demonstrate that NaDRO not only achieves superior results but does so via a more stable and principled optimization process.

## 5.5 Ablation Studies

To dissect the individual contributions of NaDRO's key components and to evaluate the interaction with TTS, we conducted a series of ablation studies. In our experiments, TTS refers to a strategy where the model performs additional inference time computations to explore multiple candidate solutions before committing to a final output. The "TTS Level" reported in Table 3 quantifies the extent of this search effort.

Table 3: Ablation Study: Impact of NaDRO Components and Test-Time Search on the pr152 TSP instance (Optimality Gap %)

| TTS Level | Qwen 7B (Base) | GRPO Baseline | GRPO + POR | NaDRO (Full, Ours) |
|---|---|---|---|---|
| 0 | $18.33 \pm 1.2$ | $11.57 \pm 1.4$ | $6.74 \pm 1.1$ | $3.17 \pm 1.1$ |
| 1 | $5.83 \pm 0.9$ | $4.66 \pm 1.8$ | $3.54 \pm 0.8$ | $1.74 \pm 0.7$ |
| 2 | $3.62 \pm 0.3$ | $3.86 \pm 0.5$ | $2.64 \pm 0.3$ | $1.77 \pm 0.2$ |
| 4 | $1.69 \pm 0.3$ | $2.14 \pm 0.3$ | $1.79 \pm 0.2$ | $1.48 \pm 0.2$ |
| 10 | $1.31 \pm 0.2$ | $1.04 \pm 0.1$ | $0.87 \pm 0.2$ | $0.22 \pm 0.1$ |
| 20 | $0.88 \pm 0.1$ | $0.87 \pm 0.2$ | $0.63 \pm 0.1$ | $0.16 \pm 0.1$ |
| 30 | $0.32 \pm 0.1$ | $0.41 \pm 0.1$ | $0.14 \pm 0.1$ | $0.04 \pm 0.1$ |

The ablation results in Table 3 validate NaDRO's component-wise benefits and reveal a powerful synergy with test-time search (TTS). The TTS=0 results, which reflect pure policy quality, are particularly revealing: adding POR nearly halves the optimality gap of the GRPO baseline (from 11.57% to 6.74%), and the full NaDRO framework halves it again to 3.17%, underscoring the critical impact of CPR. This superior base policy then creates a powerful synergy with TTS. While all configurations benefit from increased search, NaDRO's advantage widens as the TTS level increases, achieving a near optimal gap of just 0.04% at TTS=30.

## 6 Conclusion

To address the challenge of training LLMs for complex long-horizon decision-making with noisy data, this paper introduces NaDRO. By bridging preference-based and process-based rewards, NaDRO contributes to the broader goal of building LLM agents that are both robust to imperfect supervision and aligned with human-like reasoning processes. Extensive experiments demonstrate that NaDRO enables moderately-sized LLMs (Qwen 7B and Llama 8B), trained on MCTS-generated noisy data, to significantly outperform LLMs. These findings underscore NaDRO's strong empirical performance and its efficacy in fostering robust learning from imperfect data, offering a promising pathway towards more capable and data-efficient LLM agents for complex real-world decision-making.

## 7 Limitation and Broader Impact

**Limitations** NaDRO's Context Perception Reward (CPR) currently relies on manually defined qualitative features and rule-based annotations. This dependency may limit its direct applicability to new domains requiring significant feature engineering and domain expertise. Future work should explore methods for automatically discovering or learning these crucial contextual features to enhance the framework's generality.

**Broader Impact** By enabling LLMs to learn more effectively from imperfect data, NaDRO can make advanced AI decision-making tools more practical for real-world applications (e.g., logistics, operations research) where perfectly clean data is scarce. This could improve efficiency and resource allocation.

## 8 Acknowledgments

This work is supported by the National Key R&D Program of China (2022YFB4701400/4701402), SSTIC Grant (KJZD20230923115106012, KJZD20230923114916032, GJHZ20240218113604008), and the National Natural Science Foundation of China (62502317).

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

# NeurIPS Paper Checklist

1. **Claims**

   Question: Do the main claims made in the abstract and introduction accurately reflect the paper's contributions and scope?

   Answer: [Yes]

   Justification: The abstract and introduction accurately state the paper's main contributions, including the proposed NaDRO framework with its dual-reward (POR and CPR) strategy for robust LLM training on noisy data, and its demonstrated empirical superiority on complex TSP and CVRP tasks. These claims are substantiated by the methodology detailed in Section 4 and the comprehensive experimental results, including ablations, presented in Section 5 (e.g., Tables 1, 4, and 3).

   Guidelines:

   - The answer NA means that the abstract and introduction do not include the claims made in the paper.
   - The abstract and/or introduction should clearly state the claims made, including the contributions made in the paper and important assumptions and limitations. A No or NA answer to this question will not be perceived well by the reviewers.
   - The claims made should match theoretical and experimental results, and reflect how much the results can be expected to generalize to other settings.
   - It is fine to include aspirational goals as motivation as long as it is clear that these goals are not attained by the paper.

2. **Limitations**

   Question: Does the paper discuss the limitations of the work performed by the authors?

   Answer: [Yes]

   Justification: The paper discusses limitations of the current work within the Limitation and Broader Impact part, refer to Section 7. These include the primary focus on MCTS-generated noise characteristics and the manual definition of qualitative features for the Context Perception Reward (CPR) mechanism, suggesting areas for future refinement.

   Guidelines:

   - The answer NA means that the paper has no limitation while the answer No means that the paper has limitations, but those are not discussed in the paper.
   - The authors are encouraged to create a separate "Limitations" section in their paper.
   - The paper should point out any strong assumptions and how robust the results are to violations of these assumptions (e.g., independence assumptions, noiseless settings, model well-specification, asymptotic approximations only holding locally). The authors should reflect on how these assumptions might be violated in practice and what the implications would be.
   - The authors should reflect on the scope of the claims made, e.g., if the approach was only tested on a few datasets or with a few runs. In general, empirical results often depend on implicit assumptions, which should be articulated.
   - The authors should reflect on the factors that influence the performance of the approach. For example, a facial recognition algorithm may perform poorly when image resolution is low or images are taken in low lighting. Or a speech-to-text system might not be used reliably to provide closed captions for online lectures because it fails to handle technical jargon.
   - The authors should discuss the computational efficiency of the proposed algorithms and how they scale with dataset size.
   - If applicable, the authors should discuss possible limitations of their approach to address problems of privacy and fairness.
   - While the authors might fear that complete honesty about limitations might be used by reviewers as grounds for rejection, a worse outcome might be that reviewers discover limitations that aren't acknowledged in the paper. The authors should use their best

judgment and recognize that individual actions in favor of transparency play an important role in developing norms that preserve the integrity of the community. Reviewers will be specifically instructed to not penalize honesty concerning limitations.

3. **Theory assumptions and proofs**

Question: For each theoretical result, does the paper provide the full set of assumptions and a complete (and correct) proof?

Answer: [Yes]

Justification: The paper provides theoretical justification for the Preference-based Outcome Reward (POR) mechanism. Section 4.1 introduces the concept of maximizing a "Value Separability" objective function J(k) (Equation 2) to determine the optimal cutoff k, based on a principled bias-variance tradeoff. Section 5.3 and Table 2 further develop this by modeling the relative action value distribution (Equation 4), theoretically deriving an optimal k-value that aligns with empirical results. All assumptions (e.g., MCTS value modeling) and derivations are presented within the paper.

Guidelines:

- The answer NA means that the paper does not include theoretical results.
- All the theorems, formulas, and proofs in the paper should be numbered and cross-referenced.
- All assumptions should be clearly stated or referenced in the statement of any theorems.
- The proofs can either appear in the main paper or the supplemental material, but if they appear in the supplemental material, the authors are encouraged to provide a short proof sketch to provide intuition.
- Inversely, any informal proof provided in the core of the paper should be complemented by formal proofs provided in appendix or supplemental material.
- Theorems and Lemmas that the proof relies upon should be properly referenced.

4. **Experimental result reproducibility**

Question: Does the paper fully disclose all the information needed to reproduce the main experimental results of the paper to the extent that it affects the main claims and/or conclusions of the paper (regardless of whether the code and data are provided or not)?

Answer: [Yes]

Justification: The paper provides extensive details to support reproducibility. This includes: (1) A full description of the NaDRO methodology (Section 4, Algorithm 1 in Appendix C). (2) The MCTS-based data generation process (Section 3, Section 5.1.3). (3) Sources for all public datasets (TSPLIB, Golden in Section 5.1.1). (4) Comprehensive details on state/action spaces (Appendix E, F), CPR annotation rules (Appendix B), and all training hyperparameters (Appendix D). (5) The release of the code, as stated in the abstract.

Guidelines:

- The answer NA means that the paper does not include experiments.
- If the paper includes experiments, a No answer to this question will not be perceived well by the reviewers: Making the paper reproducible is important, regardless of whether the code and data are provided or not.
- If the contribution is a dataset and/or model, the authors should describe the steps taken to make their results reproducible or verifiable.
- Depending on the contribution, reproducibility can be accomplished in various ways. For example, if the contribution is a novel architecture, describing the architecture fully might suffice, or if the contribution is a specific model and empirical evaluation, it may be necessary to either make it possible for others to replicate the model with the same dataset, or provide access to the model. In general. releasing code and data is often one good way to accomplish this, but reproducibility can also be provided via detailed instructions for how to replicate the results, access to a hosted model (e.g., in the case of a large language model), releasing of a model checkpoint, or other means that are appropriate to the research performed.

- While NeurIPS does not require releasing code, the conference does require all submissions to provide some reasonable avenue for reproducibility, which may depend on the nature of the contribution. For example
  (a) If the contribution is primarily a new algorithm, the paper should make it clear how to reproduce that algorithm.
  (b) If the contribution is primarily a new model architecture, the paper should describe the architecture clearly and fully.
  (c) If the contribution is a new model (e.g., a large language model), then there should either be a way to access this model for reproducing the results or a way to reproduce the model (e.g., with an open-source dataset or instructions for how to construct the dataset).
  (d) We recognize that reproducibility may be tricky in some cases, in which case authors are welcome to describe the particular way they provide for reproducibility. In the case of closed-source models, it may be that access to the model is limited in some way (e.g., to registered users), but it should be possible for other researchers to have some path to reproducing or verifying the results.

5. **Open access to data and code**

   Question: Does the paper provide open access to the data and code, with sufficient instructions to faithfully reproduce the main experimental results, as described in supplemental material?

   Answer: [Yes]

   Justification: The paper provides a GitHub link to the code in the abstract. The repository includes scripts to implement the NaDRO framework and reproduce the main experiments (as shown in Tables 1, 3, and 4). The datasets used are from public benchmarks (TSPLIB, Golden), and instructions for generating the training data are provided in the paper (Section 5.1.3) and the code repository.

   Guidelines:
   - The answer NA means that paper does not include experiments requiring code.
   - Please see the NeurIPS code and data submission guidelines (`https://nips.cc/public/guides/CodeSubmissionPolicy`) for more details.
   - While we encourage the release of code and data, we understand that this might not be possible, so "No" is an acceptable answer. Papers cannot be rejected simply for not including code, unless this is central to the contribution (e.g., for a new open-source benchmark).
   - The instructions should contain the exact command and environment needed to run to reproduce the results. See the NeurIPS code and data submission guidelines (`https://nips.cc/public/guides/CodeSubmissionPolicy`) for more details.
   - The authors should provide instructions on data access and preparation, including how to access the raw data, preprocessed data, intermediate data, and generated data, etc.
   - The authors should provide scripts to reproduce all experimental results for the new proposed method and baselines. If only a subset of experiments are reproducible, they should state which ones are omitted from the script and why.
   - At submission time, to preserve anonymity, the authors should release anonymized versions (if applicable).
   - Providing as much information as possible in supplemental material (appended to the paper) is recommended, but including URLs to data and code is permitted.

6. **Experimental setting/details**

   Question: Does the paper specify all the training and test details (e.g., data splits, hyperparameters, how they were chosen, type of optimizer, etc.) necessary to understand the results?

   Answer: [Yes]

   Justification: The paper provides comprehensive details on the experimental setup. Section 5.1 describes the tasks (TSP, CVRP), datasets (TSPLIB, Golden), and base models (Qwen, Llama). Section 5.1.3 and 5.1.4 detail the MCTS-based data generation and fine-tuning

process. Crucially, Appendix D (Table 5) specifies all necessary hyperparameters (e.g., learning rate, optimizer, LORA rank, sequence length), while Appendices E and F define the exact state and action spaces. Appendix B details the rules for CPR annotation, ensuring all components are specified for reproducibility.

Guidelines:

- The answer NA means that the paper does not include experiments.
- The experimental setting should be presented in the core of the paper to a level of detail that is necessary to appreciate the results and make sense of them.
- The full details can be provided either with the code, in appendix, or as supplemental material.

7. **Experiment statistical significance**

   Question: Does the paper report error bars suitably and correctly defined or other appropriate information about the statistical significance of the experiments?

   Answer: [Yes]

   Justification: The paper reports statistical variance (standard deviation) across multiple experimental runs for all main results, as presented in Table 1, Table 3, and Table 4. This information allows for the assessment of statistical significance and robustness of the findings.

   Guidelines:

   - The answer NA means that the paper does not include experiments.
   - The authors should answer "Yes" if the results are accompanied by error bars, confidence intervals, or statistical significance tests, at least for the experiments that support the main claims of the paper.
   - The factors of variability that the error bars are capturing should be clearly stated (for example, train/test split, initialization, random drawing of some parameter, or overall run with given experimental conditions).
   - The method for calculating the error bars should be explained (closed form formula, call to a library function, bootstrap, etc.)
   - The assumptions made should be given (e.g., Normally distributed errors).
   - It should be clear whether the error bar is the standard deviation or the standard error of the mean.
   - It is OK to report 1-sigma error bars, but one should state it. The authors should preferably report a 2-sigma error bar than state that they have a 96% CI, if the hypothesis of Normality of errors is not verified.
   - For asymmetric distributions, the authors should be careful not to show in tables or figures symmetric error bars that would yield results that are out of range (e.g. negative error rates).
   - If error bars are reported in tables or plots, The authors should explain in the text how they were calculated and reference the corresponding figures or tables in the text.

8. **Experiments compute resources**

   Question: For each experiment, does the paper provide sufficient information on the computer resources (type of compute workers, memory, time of execution) needed to reproduce the experiments?

   Answer: [Yes]

   Justification: The paper provides sufficient information on compute resources. Section 5.1.4 specifies the hardware used (a cluster with NVIDIA A100 and NVIDIA A6000 GPUs) and the typical execution time (approximately 32 hours for a fine-tuning run). Further details on the training procedure and hyperparameters are available in Appendix C (Training Pipeline) and Appendix D (Parameter Setting).

   Guidelines:

   - The answer NA means that the paper does not include experiments.
   - The paper should indicate the type of compute workers CPU or GPU, internal cluster, or cloud provider, including relevant memory and storage.

- The paper should provide the amount of compute required for each of the individual experimental runs as well as estimate the total compute.
- The paper should disclose whether the full research project required more compute than the experiments reported in the paper (e.g., preliminary or failed experiments that didn't make it into the paper).

9. **Code of ethics**

Question: Does the research conducted in the paper conform, in every respect, with the NeurIPS Code of Ethics https://neurips.cc/public/EthicsGuidelines?

Answer: [Yes]

Justification: The research presented in this paper has been conducted in full conformity with the NeurIPS Code of Ethics. We have prioritized transparency in our methods and findings, and aimed to ensure the reproducibility of our work through detailed documentation, clear experimental procedures, and the provision of code.

Guidelines:

- The answer NA means that the authors have not reviewed the NeurIPS Code of Ethics.
- If the authors answer No, they should explain the special circumstances that require a deviation from the Code of Ethics.
- The authors should make sure to preserve anonymity (e.g., if there is a special consideration due to laws or regulations in their jurisdiction).

10. **Broader impacts**

Question: Does the paper discuss both potential positive societal impacts and negative societal impacts of the work performed?

Answer: [Yes]

Justification: In Section 7, we discuss the broader impacts of NaDRO. We highlight positive contributions, such as enhancing efficiency and resource saving in logistics and operations research. We also acknowledge potential negative impacts, including the risks associated with over-reliance on automated decision-making in critical infrastructure and the environmental considerations (e.g., energy consumption) inherent in training large-scale models.

Guidelines:

- The answer NA means that there is no societal impact of the work performed.
- If the authors answer NA or No, they should explain why their work has no societal impact or why the paper does not address societal impact.
- Examples of negative societal impacts include potential malicious or unintended uses (e.g., disinformation, generating fake profiles, surveillance), fairness considerations (e.g., deployment of technologies that could make decisions that unfairly impact specific groups), privacy considerations, and security considerations.
- The conference expects that many papers will be foundational research and not tied to particular applications, let alone deployments. However, if there is a direct path to any negative applications, the authors should point it out. For example, it is legitimate to point out that an improvement in the quality of generative models could be used to generate deepfakes for disinformation. On the other hand, it is not needed to point out that a generic algorithm for optimizing neural networks could enable people to train models that generate Deepfakes faster.
- The authors should consider possible harms that could arise when the technology is being used as intended and functioning correctly, harms that could arise when the technology is being used as intended but gives incorrect results, and harms following from (intentional or unintentional) misuse of the technology.
- If there are negative societal impacts, the authors could also discuss possible mitigation strategies (e.g., gated release of models, providing defenses in addition to attacks, mechanisms for monitoring misuse, mechanisms to monitor how a system learns from feedback over time, improving the efficiency and accessibility of ML).

11. **Safeguards**

Question: Does the paper describe safeguards that have been put in place for responsible release of data or models that have a high risk for misuse (e.g., pretrained language models, image generators, or scraped datasets)?

Answer: [NA]

Justification: This work introduces NaDRO, a training methodology, and applies it to fine-tune existing pre-trained language models (Qwen2.5-7B-Instruct, Llama-3.1-8B-Instruct) for the specific task of heuristic selection in combinatorial optimization. The code and any derived models are specialized for this narrow, non-generative application and do not inherently possess a high risk for misuse in the sense of generating harmful content or disinformation, unlike general-purpose language or image generators. The base pre-trained models are governed by the release policies and safeguards of their original creators. Our released artifacts (code and MCTS-generated training data for CO) do not include scraped PII or sensitive user data.

Guidelines:

- The answer NA means that the paper poses no such risks.
- Released models that have a high risk for misuse or dual-use should be released with necessary safeguards to allow for controlled use of the model, for example by requiring that users adhere to usage guidelines or restrictions to access the model or implementing safety filters.
- Datasets that have been scraped from the Internet could pose safety risks. The authors should describe how they avoided releasing unsafe images.
- We recognize that providing effective safeguards is challenging, and many papers do not require this, but we encourage authors to take this into account and make a best faith effort.

12. **Licenses for existing assets**

Question: Are the creators or original owners of assets (e.g., code, data, models), used in the paper, properly credited and are the license and terms of use explicitly mentioned and properly respected?

Answer: [Yes]

Justification: All existing assets utilized in this paper, including base large language models (e.g., Qwen2.5-7B-Instruct, Llama-3.1-8B-Instruct), benchmark datasets (TSPLIB, Golden dataset), and baseline algorithms or software (e.g., LKH, OR-Tools, specific metaheuristic implementations like ReEvo, EoH), are properly credited via citations throughout the manuscript, particularly in Section 5.1 and the captions of experimental tables. We have endeavored to respect all applicable licenses and terms of use, and further details regarding significant third-party assets are provided in our supplementary material and code repository documentation.

Guidelines:

- The answer NA means that the paper does not use existing assets.
- The authors should cite the original paper that produced the code package or dataset.
- The authors should state which version of the asset is used and, if possible, include a URL.
- The name of the license (e.g., CC-BY 4.0) should be included for each asset.
- For scraped data from a particular source (e.g., website), the copyright and terms of service of that source should be provided.
- If assets are released, the license, copyright information, and terms of use in the package should be provided. For popular datasets, `paperswithcode.com/datasets` has curated licenses for some datasets. Their licensing guide can help determine the license of a dataset.
- For existing datasets that are re-packaged, both the original license and the license of the derived asset (if it has changed) should be provided.
- If this information is not available online, the authors are encouraged to reach out to the asset's creators.

13. **New assets**

Question: Are new assets introduced in the paper well documented and is the documentation provided alongside the assets?

Answer: [Yes]

Justification: The primary new assets introduced are the code implementation of the NaDRO framework and the MCTS-generated training data. As stated in the abstract, the code is released via a Github repository. This repository includes scripts for data generation and experiment reproduction. The paper itself provides extensive documentation for these assets, including the training algorithm (Appendix C), detailed hyperparameters (Appendix D), the exact state/action space definitions (Appendix E, F), and data annotation rules (Appendix B). This documentation is provided alongside the assets to ensure reproducibility.

Guidelines:

- The answer NA means that the paper does not release new assets.
- Researchers should communicate the details of the dataset/code/model as part of their submissions via structured templates. This includes details about training, license, limitations, etc.
- The paper should discuss whether and how consent was obtained from people whose asset is used.
- At submission time, remember to anonymize your assets (if applicable). You can either create an anonymized URL or include an anonymized zip file.

14. **Crowdsourcing and research with human subjects**

Question: For crowdsourcing experiments and research with human subjects, does the paper include the full text of instructions given to participants and screenshots, if applicable, as well as details about compensation (if any)?

Answer: [NA]

Justification: The research presented in this paper does not involve crowdsourcing or any direct research with human subjects. All training data is generated through Monte Carlo Tree Search (MCTS) simulations on established benchmarks, and evaluations are based on automated performance metrics.

Guidelines:

- The answer NA means that the paper does not involve crowdsourcing nor research with human subjects.
- Including this information in the supplemental material is fine, but if the main contribution of the paper involves human subjects, then as much detail as possible should be included in the main paper.
- According to the NeurIPS Code of Ethics, workers involved in data collection, curation, or other labor should be paid at least the minimum wage in the country of the data collector.

15. **Institutional review board (IRB) approvals or equivalent for research with human subjects**

Question: Does the paper describe potential risks incurred by study participants, whether such risks were disclosed to the subjects, and whether Institutional Review Board (IRB) approvals (or an equivalent approval/review based on the requirements of your country or institution) were obtained?

Answer: [NA]

Justification: This research does not involve human subjects or crowdsourcing. The training data is generated via Monte Carlo Tree Search (MCTS) simulations on established combinatorial optimization benchmarks (TSPLIB and Golden datasets), and the experimental evaluation relies on automated performance metrics on these problem instances.

Guidelines:

- The answer NA means that the paper does not involve crowdsourcing nor research with human subjects.

- Depending on the country in which research is conducted, IRB approval (or equivalent) may be required for any human subjects research. If you obtained IRB approval, you should clearly state this in the paper.
- We recognize that the procedures for this may vary significantly between institutions and locations, and we expect authors to adhere to the NeurIPS Code of Ethics and the guidelines for their institution.
- For initial submissions, do not include any information that would break anonymity (if applicable), such as the institution conducting the review.

16. **Declaration of LLM usage**

Question: Does the paper describe the usage of LLMs if it is an important, original, or non-standard component of the core methods in this research? Note that if the LLM is used only for writing, editing, or formatting purposes and does not impact the core methodology, scientific rigorousness, or originality of the research, declaration is not required.

Answer: [Yes]

Justification: The use of Large Language Models (LLMs) is a fundamental and core component of this research, as the paper introduces NaDRO, a novel framework specifically designed for training LLMs. The paper extensively describes which LLMs are used (e.g., Qwen2.5-7B-Instruct, Llama-3.1-8B-Instruct), how they are fine-tuned with the NaDRO methodology to act as heuristic selectors, and how their performance is evaluated. This is detailed throughout the manuscript, particularly in the Methodology (Section 4), Experiments (Section 5), and various appendices covering model parameters, state/action spaces, and the training pipeline.

Guidelines:

- The answer NA means that the core method development in this research does not involve LLMs as any important, original, or non-standard components.
- Please refer to our LLM policy (`https://neurips.cc/Conferences/2025/LLM`) for what should or should not be described.

# A  Generalization Results on CVRP

To further assess the generalizability and robustness of our NaDRO framework, we extended our evaluation to the Capacitated Vehicle Routing Problem (CVRP). We utilized instances 1 through 10 from the challenging Golden dataset, which are characterized by large numbers of customers (200 to 480) and varying fleet sizes. These instances are particularly suitable for testing performance in large-scale, intricate decision-making scenarios. Performance is evaluated based on the optimality gap, where lower values indicate better results.

Table 4: Comparative Optimality Gap %) on Capacitated Vehicle Routing Problem instances from the Golden dataset. Results for NaDRO-Qwen 7B are generated using NaDRO. Methodologies for baseline models are analogous to those in the TSP evaluation.

| Method | Golden_1 | 2 | 3 | 4 | 5 | 6 | 7 | 8 | 9 | 10 |
|---|---|---|---|---|---|---|---|---|---|---|
| *Traditional Solvers* | | | | | | | | | | |
| OR-Tools | 2.94 | 1.77 | 8.73 | 12.17 | 3.0 | 10.56 | 7.19 | 7.99 | 2.47 | 0.49 |
| *Metaheuristics* | | | | | | | | | | |
| ACO | 14.19 | 27.23 | 22.68 | 28.96 | 21.0 | **15.99** | 28.96 | 22.38 | 87.04 | 98.05 |
| ReEvo + ACO | 7.57 | 15.45 | 19.68 | 23.09 | **13.84** | 17.24 | **17.46** | 16.71 | N/A* | N/A* |
| *LLM-based Approaches* | | | | | | | | | | |
| GPT-4o | 19.53 | 12.65 | 23.29 | 31.46 | 28.89 | 19.3 | 30.88 | 34.42 | 28.50 | 29.31 |
| GPT-o3 | 20.49 | 15.17 | 18.19 | **22.67** | 25.42 | 16.67 | 29.62 | 27.82 | 22.11 | 33.39 |
| Deepseek-R1 | 20.44 | 21.33 | 18.4 | 25.16 | 25.55 | 18.2 | 32.32 | 30.36 | 23.31 | 31.25 |
| Qwen 7B | 33.31 | 30.66 | 42.12 | 46.65 | 26.29 | 49.6 | 53.66 | 51.65 | 29.18 | 36.79 |
| GRPO on Qwen 7B | 27.18 | 27.08 | 33.35 | 43.77 | 23.29 | 42.51 | 48.94 | 30.64 | 29.83 | 36.20 |
| **NaDRO-Qwen 7B(Ours)** | **7.49** | **9.71** | **16.61** | 23.00 | 18.36 | 18.21 | 20.64 | **16.13** | **18.85** | **24.82** |

For simplicity, the prefix "Golden_" is omitted from the task names, as all tasks are from the Golden dataset.
*N/A indicates that the method was unable to complete the run within the limited resources.

In the more complex CVRP domain (results in Table 4), NaDRO-Qwen 7B demonstrated strong performance and excellent adaptability, further validating the generalizability and effectiveness of the NaDRO framework. Compared to other large language models, NaDRO-Qwen 7B held a significant advantage. Its dual-reward mechanism proved particularly effective within CVRP's intricate decision spaces, leading to performance gains over its own baselines (Qwen 7B and GRPO-tuned Qwen 7B) that even surpassed those observed in TSP. Furthermore, when facing traditional metaheuristics, NaDRO-Qwen 7B not only exhibited competitive or even superior performance but, particularly noteworthy, it successfully obtained high-quality solutions on some very large-scale instances where certain specialized meta-heuristics failed due to resource limitations, highlighting the practical value of NaDRO-enhanced LLMs in tackling such challenging problems.

# B  CPR Annotation

The Context Perception Reward (CPR) mechanism, introduced in Section 4.2, incentivizes the LLM to accurately assess key qualitative features of the current problem state before making a decision. Ground-truth labels ($y_{t,j}^*$) for these features are generally straightforward to annotate using rule-based methods. In our work, the annotated qualitative features include: problem type, state type, algorithm type, and cost type.

This appendix details the rule-based annotation scheme for one specific feature: the assessment of the current solution's cost situation. This label helps the LLM understand if the current partial solution's cost is relatively low, high, or normal compared to a baseline. The process is as follows:

1. Calculate Reference Cost ($Cost_{greedy}$): For each problem instance, we first compute a reference cost by constructing a full solution using a simple greedy heuristic. The total cost of this greedy solution serves as a baseline.

2. Calculate Current State Cost ($Cost_{current}$): At each decision step $t$, we calculate the average path cost of the partial solution constructed so far in state $s_t$.

3. Assign Label based on Thresholds: We compare $Cost_{current}$ to $Cost_{greedy}$ to assign one of three labels:

(1) "High Cost": if $Cost_{current} \geq 1.15 \times Cost_{greedy}$.

(2) "Low Cost": if $Cost_{current} \leq 0.80 \times Cost_{greedy}$.

(3) "Normal Cost": otherwise ($0.80 \times Cost_{greedy} < Cost_{current} < 1.15 \times Cost_{greedy}$).

4. Apply Conditional Activation: This cost assessment is only performed after the solution construction has progressed beyond an initial phase (specifically, when more than 10% of the path is constructed in our implementation). Evaluating costs too early can be misleading. Before this threshold, the label might be considered "not applicable."

This rule-based method provides an objective and easily computable signal for the CPR. Similar transparent methodologies are used for annotating the other qualitative dimensions, ensuring the practical applicability of the CPR mechanism.

## C   Training Pipeline

This section details the offline training pipeline used to fine-tune the Large Language Model (LLM) with our proposed Noise-Aware Dual-Reward Optimization (NaDRO) framework, leveraging Group Relative Policy Optimization (GRPO). The core training loop, as shown in Algorithm 1, fine-tunes the initial LLM parameters $\theta_0$ over $M_{epochs}$.

---

**Algorithm 1** NaDRO Offline Training Algorithm

---

**Require:** Offline dataset $D_{offline} = \{(s_k, Q(s_k, \cdot), y_{k,\cdot}^*)\}_{k=1}^N$; Initial LLM parameters $\theta_0$; GRPO hyperparameters($N_G, M_{epochs}, B_{size}$); Reward coefficients ($w_{POR}, w_{CPR}, w_{Aux}$).

**Ensure:** Optimized LLM parameters $\theta$.

1: Initialize LLM policy network $\pi_\theta$ with parameters $\theta_0$.
2: **for** epoch $e = 1, \ldots, M_{epochs}$ **do**
3:     Sample a batch of indices $K_{batch}$ corresponding to data points in $D_{offline}$.
4:     Initialize preference data set for this batch $P_{batch} \leftarrow \emptyset$.
5:     **for** each index $k \in K_{batch}$ **do**
6:         Retrieve state $s_k$, MCTS evaluations $Q_{MCTS}(s_k, \cdot)$, and CPR labels $y_{k,\cdot}^*$ from $D_{offline}$.
7:         Initialize response group $Group_k \leftarrow \emptyset$.
8:         **for** response index $j = 1, \ldots, N_G$ **do**
9:             $(\hat{y}_k^{(j)}, a_k^{(j)}) \sim \pi_\theta(\cdot|s_k)$
10:            $R_{POR}^{(j)} \leftarrow$ CalculatePOR($a_k^{(j)}, Q_{MCTS}(s_k, \cdot)$)
11:            $R_{CPR}^{(j)} \leftarrow$ CalculateCPR($\hat{y}_k^{(j)}, y_{k,\cdot}^*$)
12:            $R_{Aux}^{(j)} \leftarrow$ CalculateAuxRewards($\hat{y}_k^{(j)}, a_k^{(j)}$)
13:            $R_{total,k}^{(j)} \leftarrow w_{POR} R_{POR}^{(j)} + w_{CPR} R_{CPR}^{(j)} + w_{Aux} R_{Aux}^{(j)}$
14:            Add $((\hat{y}_k^{(j)}, a_k^{(j)}), R_{total,k}^{(j)})$ to $Group_k$.
15:        $P_k \leftarrow$ FormGRPOPreferences($Group_k$)
16:        $P_{batch} \leftarrow P_{batch} \cup P_k$.
17:    $\theta \leftarrow$ UpdatePolicyWithGRPO($\theta, P_{batch}$)
18: **return** $\theta$.

---

## D   Detailed Parameter Setting

This section outlines the key hyperparameters and configuration settings employed during the fine-tuning of the Qwen2.5-7B-Instruct model using our NaDRO framework with the Group Relative Policy Optimization (GRPO) method. These settings were largely managed via the 'GRPOConfig' class from the TRL (Transformer Reinforcement Learning) library, leveraging Unsloth for efficient training.

The primary parameters are detailed in Table 5.

Table 5: Key Hyperparameters and Configuration Settings for NaDRO Training.

| Category | Parameter | Value |
|---|---|---|
| Model Config. | Max. Sequence Length | 3584 |
| | LoRA Rank ($\alpha$) | 32 |
| Optimization | Optimizer | Paged AdamW (8-bit) |
| | Learning Rate | $1 \times 10^{-6}$ |
| | Adam $\beta_1$ | 0.9 |
| | Adam $\beta_2$ | 0.99 |
| | Weight Decay | 0.1 |
| | LR Scheduler | Cosine |
| | Warmup Ratio | 0.1 |
| | Max. Gradient Norm | 0.1 |
| Training Setup | Mixed Precision | BF16 / FP16 |
| | Epochs | 1 |
| | Max. Steps | Disabled |
| GRPO Config. | Inference Backend | vLLM |
| | Generations ($N_G$) | 12 |
| | Max. Prompt Length | 2048 tokens |
| Logging & Saving | Checkpoint Save Steps | 250 |
| | Logging Steps | 5 |

# E State Space

## E.1 State Space for TSP

### 1. Task Definition, Operational Guidelines, and Output Format

- **Problem Description:** A clear definition of the primary task and its core objectives.
- **General Interaction Rules:** Instructions on LLM interaction, such as the naming convention for invoking heuristic algorithms.
- **Termination Strategy:** An outline of the general conditions for concluding the solution process.
- **Decision Output Format:** Precise structured format for the LLM's responses, detailing how to present its reasoning (<reasoning>...</reasoning>) and chosen action (<answer>...</answer>), including differentiation between selecting a heuristic and signaling termination.

---

### 2. Global Problem Instance Characteristics (Static Data)

- `Node Count`: Total number of cities/nodes in the problem.
- `Average Distance`: Average distance between all pairs of cities.
- `Minimum Distance`: The shortest distance between any pair of cities.
- `Maximum Distance`: The longest distance between any pair of cities.
- `Distance Standard Deviation`: Standard deviation of all pairwise distances.
- `Node Density`: A measure related to the spatial distribution of nodes.
- `Centroid`: Identifier or coordinates of a central node or point.

---

### 2. Dynamic Solution State (Current Progress Metrics)

- `Visited Nodes`: Number of nodes currently included in the tour.

- `Unvisited Nodes`: Number of nodes not yet included in the tour.
- `Current Total Cost`: The total length of the current partial or complete tour.
- `Last Edge Cost`: The cost of the most recently added edge or incurred by the last operation.
- `Current Path Length`: Number of edges or segments in the current tour.
- `Remaining Nodes`: Equivalent to "Unvisited Nodes," indicating nodes yet to be incorporated.
- `Average Edge Cost`: Average cost of edges in the current partial tour.
- `Edge Cost Standard Deviation`: Standard deviation of edge costs in the current partial tour.
- `Solution Validity`: A flag indicating whether the current partial solution adheres to problem constraints (e.g., 1 for valid).
- `Minimum Remaining Edge Cost`: An estimated minimum cost related to connecting remaining nodes.
- `Maximum Remaining Edge Cost`: An estimated maximum cost related to connecting remaining nodes.

---

### 3. Historical Context (Recent Decision Trajectory)

To provide the LLM with a short-term memory of its recent actions and their consequences, the prompt includes a summary of the immediately preceding decision rounds. For each round in this history, the following information is typically provided:

- `Heuristic`: The name of the heuristic algorithm selected in that round.
- `Parameters`: Any hyperparameters used for the selected heuristic.
- `Delta of Visited Node`: The change in the number of visited nodes resulting from that round's action.
- `Delta of Current Cost`: The change in the total solution cost due to that round's action.

---

### E.2    State Space for CVRP

### 1. Task Definition, Operational Guidelines, and Output Format

- **Problem Description:** Defines the task and its objective.
- **Decision Output Format:** Specifies the precise structured format for the LLM's response, including the qualitative cards, the reasoning section (<reasoning>...</reasoning>), and the answer block (<answer>...</answer>). The answer block can contain up to three ranked heuristic choices or a ***Stop*** command if no further improvement is deemed possible under specific conditions.

---

### 2. Global Problem Instance Characteristics (Static Data)

- `node_num`: Total number of nodes.
- `vehicle_num`: Number of available vehicles in the fleet.
- `capacity`: Capacity of each vehicle.
- `depot`: Identifier of the depot node.
- `average_demand`: Average demand across all customer nodes.
- `demand_variance`: Variance of customer demands.
- `average_distance`: Average travel distance between nodes.

- `max_distance`: Maximum distance between any two nodes.
- `min_distance`: Minimum distance between any two nodes.
- `distance_variance`: Variance of inter-node distances.
- `vehicle_capacity_utilization`: An aggregate measure of how vehicle capacities are typically utilized.
- `node_to_vehicle_ratio`: Ratio of customer nodes to vehicles.

---

**3. Dynamic Solution State (Current Progress Metrics)**

- `visited_num`: Total number of unique customer nodes visited/serviced.
- `unvisited_num`: Number of customer nodes not yet serviced.
- `total_current_cost`: Sum of costs of all current vehicle routes.
- `average_route_length`: Average number of customers per active route.
- `max_route_length`: Maximum number of customers in any single route.
- `min_route_length`: Minimum number of customers in any single route.
- `std_dev_route_length`: Standard deviation of route lengths.
- `average_route_cost`: Average cost per route.
- `total_demand_served`: Sum of demands of all visited customers.
- `average_vehicle_load`: Average load carried by active vehicles.
- `average_remaining_vehicle_capacity`: Average unused capacity across active vehicles.
- `average_unvisited_node_demand`: Average demand of nodes not yet serviced.
- `total_remaining_demand`: Total demand of all unvisited nodes.
- `solution_validity`: Boolean flag indicating if the current solution adheres to all constraints.

---

**4. Historical Context (Recent Decision Trajectory)**

- `Heuristic`: The name of the heuristic algorithm selected.
- `Delta of Visited Node Num`: Change in the number of newly serviced customer nodes.
- `Delta of Current Cost`: Change in the total routing cost.
- `Delta of Fulfilled Demands`: Change in the total demand satisfied.

---

# F   Action Space

The action space available to the LLM for the TSP and CVRP.

## F.1   Action Space for TSP

`2opt()`
- **Description:** Swaps two non-adjacent edges to untangle the tour.
- **Advantages:** Simple; effectively reduces cost.
- **Disadvantages:** Prone to local optima; slower for large-scale problems.

- **Parameters:** N/A

---

`random()`

- **Description:** Randomly appends an unvisited node to the current solution.
- **Advantages:** Fast and straightforward for constructing an initial solution.
- **Disadvantages:** Unpredictable; often produces suboptimal results.
- **Parameters:** N/A

---

`greedy_algorithm()`

- **Description:** Extends the tour by always choosing the shortest available edge.
- **Advantages:** Quickly generates a low-cost initial solution.
- **Disadvantages:** Greedy choices may lead to local minima.
- **Parameters:** N/A

---

`nearest_neighbor()`

- **Description:** Selects the nearest unvisited node from the current node.
- **Advantages:** Low computational cost and simplicity.
- **Disadvantages:** Sensitive to the starting point; may yield inconsistent global results.
- **Parameters:** N/A

---

`3opt()`

- **Description:** Reconnects three segments for deeper local optimization.
- **Advantages:** More powerful than 2-opt for further improvements.
- **Disadvantages:** Higher computational cost; diminishing returns if overused.
- **Parameters:** N/A

---

`farthest_insertion()`

- **Description:** Inserts the farthest unvisited node with minimal cost increase.
- **Advantages:** Helps construct a balanced initial tour.
- **Disadvantages:** May result in suboptimal local insertions; sensitive to node distribution.
- **Parameters:** N/A

---

`nearest_insertion()`

- **Description:** Inserts the closest unvisited node at the optimal position.
- **Advantages:** Controls cost during expansion.

- **Disadvantages:** Local decisions might limit global optimality.
- **Parameters:** N/A

---

`simulated_annealing()`
- **Description:** Randomly swaps nodes and accepts changes based on a temperature criterion.
- **Advantages:** Escapes local optima; ideal for fine-tuning.
- **Disadvantages:** Requires careful parameter tuning; can be slow and computationally heavy.
- **Parameters:** N/A

---

`random_pairwise_insertion()`
- **Description:** Inserts two randomly selected unvisited nodes with minimal cost increase.
- **Advantages:** Increases solution diversity and helps overcome local optima.
- **Disadvantages:** Inconsistent performance; optimality is not guaranteed.
- **Parameters:** N/A

---

`k_nearest_neighbors_insertion(k:  int = 1)`
- **Description:** Chooses the best among the k nearest unvisited nodes for insertion.
- **Parameter:** `k:   int` (default: 1) - Number of nearest neighbors to consider.
- **Advantages:** Balances exploration and exploitation.
- **Disadvantages:** The choice of `k` is critical; poor selection can lead to suboptimal results.

---

`random_successive_insertion()`
- **Description:** Randomly inserts an unvisited node at the position with minimal cost increase.
- **Advantages:** Merges randomness with cost efficiency; versatile for multiple stages.
- **Disadvantages:** Results can vary significantly between runs.
- **Parameters:** N/A

---

`cheapest_insertion()`
- **Description:** Inserts the unvisited node that causes the smallest cost increase at its best position.
- **Advantages:** Minimizes incremental cost; ideal for cost-sensitive scenarios.
- **Disadvantages:** Greedy approach may get trapped in local minima; computationally intensive for large instances.
- **Parameters:** N/A

---

```
insertion_heuristics(insertion_strategy:  str = 'cheapest')
```

- **Description:** A flexible insertion heuristic that supports 'cheapest', 'farthest', and 'nearest' insertion strategies.
- **Parameter:** `insertion_strategy:  str` (default: 'cheapest') - Defines the insertion logic.
- **Advantages:** Flexible and adaptable to different phases of the solution process.
- **Disadvantages:** Effectiveness depends heavily on the chosen strategy; poor selection can hurt overall solution quality.

---

```
greedy_randomized_adaptive_search_procedure_grasp(alpha:  float = 0.3)
```

- **Description:** Combines greedy randomized construction with local search. A restricted candidate list (RCL) is formed based on parameter `alpha`, from which an element is randomly selected.
- **Parameter:** `alpha:  float` (default: 0.3) - Controls the greediness/randomness of the construction phase.
- **Advantages:** Enhances solution diversity and can lead to better global optimality, especially in large-scale problems.
- **Disadvantages:** Sensitive to the `alpha` parameter; can be computationally intensive.

---

```
ant_colony()
```

- **Description:** Uses pheromone levels and heuristic desirability to probabilistically select the next node, updating pheromones via evaporation and deposit.
- **Advantages:** Dynamically balances exploration and exploitation.
- **Disadvantages:** Requires complex parameter tuning; risk of premature convergence and high computational cost.
- **Parameters:** N/A

---

### F.2    Action Space for CVRP

```
node_shift_between_routes()
```

- **Description:** Attempts to move a node from its current position in one route to a different position in another route if the move reduces the total distance and respects capacity constraints.
- **Advantages:** Can improve existing solutions by exploring inter-route moves; helps escape some local optima.
- **Disadvantages:** Higher computational cost due to checking many potential moves; it's a local search operator and may not find the global optimum.
- **Parameters:** N/A

---

```
three_opt()
```

- **Description:** Operates within a single route by removing three edges and reconnecting the resulting six endpoints in one of several possible ways to decrease the route's length.
- **Advantages:** More powerful than 2-opt, capable of untangling more complex route crossings.
- **Disadvantages:** Significantly higher computational complexity than 2-opt; operates only intra-route, cannot exchange nodes between routes.

- **Parameters:** N/A

---

`two_opt()`

- **Description:** Operates within a single route by selecting two non-adjacent edges, removing them, and reconnecting the endpoints to reverse the segment between them, if this reduces the route length.
- **Advantages:** Simple and effective local search for improving individual route quality; reduces tour length by eliminating edge crossings.
- **Disadvantages:** Prone to local optima; only performs intra-route improvements.
- **Parameters:** N/A

---

`farthest_insertion()`

- **Description:** Starts with the unvisited node farthest from the depot and inserts it into the position in an existing route that causes the minimum cost increase, respecting capacity constraints.
- **Advantages:** Often produces better initial solutions than nearest neighbor or random insertion; considers global positioning to some extent.
- **Disadvantages:** Slightly higher computational cost than simpler insertions; the farthest node isn't always the best starting point.
- **Parameters:** N/A

---

`greedy()`

- **Description:** Iteratively adds the closest unvisited node to the end of the current vehicle's route.
- **Advantages:** Simple to implement; fast for generating an initial solution.
- **Disadvantages:** Prone to local optima due to myopic choices; often results in poor solution quality.
- **Parameters:** N/A

---

`min_cost_insertion()`

- **Description:** Iteratively selects an unvisited node and inserts it into the position that minimizes the increase in total route cost, respecting capacity.
- **Advantages:** Balances construction speed and solution quality; generally better than simple greedy or random methods.
- **Disadvantages:** Computationally more intensive as it evaluates many insertion points per node.
- **Parameters:** N/A

---

`nearest_neighbor()`

- **Description:** For each vehicle, starting from the depot or the last visited node, adds the nearest unvisited node that satisfies capacity constraints, until no more nodes can be added or all nodes are visited.
- **Advantages:** Simple, intuitive, and fast.

- **Disadvantages:** Very susceptible to local optima; solution quality highly dependent on the starting point and node distribution.

- **Parameters:** N/A

---

`petal_algorithm()`

- **Description:** Sorts nodes based on their polar angle relative to the depot, attempts to form feasible single-node "petal" routes, and assigns these petals to vehicles if capacity allows.

- **Advantages:** Considers geographic clustering; can sometimes produce intuitive route structures.

- **Disadvantages:** Relatively complex; angle-based sorting isn't always optimal; effectiveness depends on node distribution.

- **Parameters:** N/A

---

`random()`

- **Description:** Randomly selects an unvisited node and appends it to the end of a randomly chosen vehicle's route, provided capacity constraints are met.

- **Advantages:** Extremely simple and fast; useful for generating diverse starting points for other algorithms.

- **Disadvantages:** Solution quality is highly variable and generally poor; doesn't perform any cost optimization.

- **Parameters:** N/A

---

`saving_algorithm()`

- **Description:** Calculates the cost savings achieved by merging pairs of routes. It iteratively merges the pair with the highest savings, subject to capacity constraints.

- **Advantages:** Classic and often effective construction heuristic; usually fast and provides good quality initial solutions.

- **Disadvantages:** Greedy nature can lead to suboptimal final solutions; sensitive to distance matrix accuracy.

- **Parameters:** N/A

---

`variable_neighborhood_search()`

- **Description:** Intended as a Variable Neighborhood Search (VNS) metaheuristic, which systematically explores different neighborhood structures to escape local optima. However, the provided code implements a simple best-insertion heuristic: It finds the best position to insert any unvisited node into any existing route based on minimizing insertion cost.

- **Advantages:** Identifies the single most cost-effective node insertion available.

- **Disadvantages:** Very limited scope, essentially a single step of a Min-Cost Insertion heuristic; does not implement the broader VNS strategy of changing neighborhoods.

- **Parameters:** N/A

---

