# OpenReview forum: "NaDRO: Leveraging Dual-Reward Strategies for LLMs Training on Noisy Data"
_NeurIPS.cc/2025/Conference — NeurIPS 2025 poster_

### Official Review · Reviewer_tJhr · 2025-06-25

**Clarity:** 3
**Significance:** 2
**Originality:** 3
**Rating:** 4
**Confidence:** 3

**Summary:**

This paper proposes a novel approach to help LLM learn effectively from noisy or incomplete data. The proposed algorithm, NaDRO, which is built on dual rewards: preference-based outcome reward and context perception reward, synergistically optimizes learning signals from both outcome preference and intermediate reasoning process. The authors validate the empirical performance on two specific problems: travelling salesman problem and capacitated vehicle routing problem.

**Questions:**

Questions:
1. While the NaDRO outperforms other LLM-based methods on MCTS-generated data, the conclusion in claiming offering a promising pathway towards real-world noisy data handling is not fully justified by current experiments. I would suggest adding a brief discussion on what types of real-world noise might be relevant and how NaDRO could be adapted or expected to handle them.
2. The paper adopts fixed thresholds (1.15 and 0.8) for labeling high and low cost in CPR annotation, but it’s unclear how these specific values were determined.
3. The empirical evaluation does not demonstrate a clear advantage over classical baselines such as LKH in TSP, which raises concerns about the practical relevance of using LLMs for these tasks. What is the advantage of LLM-based methods in such tasks and when would such method be preferable to strong classical solvers?

**Ethical Concerns:**

["NO or VERY MINOR ethics concerns only"]

**Final Justification:**

Thank you to the authors for the thorough and thoughtful rebuttal. As the main concerns I raised have now been addressed, I am updating my overall score to 4.

**Limitations:**

See Weaknesses and Questions.

**Paper Formatting Concerns:**

No Concerns.

**Quality:**

3

**Strengths And Weaknesses:**

Strengths:
The paper is well-written, with a clear and logical flow that makes it easy to follow. The methodology is explained in detail, providing a solid understanding of the approach. The ablation study shows the importance of each component clearly.

Weaknesses:
1. The ablation studies clearly demonstrate that the dual-reward strategy improves performance. However, it is better for the authors to more explicitly articulate the intuition or theoretical basis for combining these specific rewards.
2. Given that the paper does not provide a theoretical analysis for NaDRO, the empirical results become only evidence supporting the effectiveness of the method. Therefore, it is better to add even a minimal variance analysis across different random seeds to demonstrate that the observed performance improvements are robust.
3. In the experiments, the comparison methods are appropriately cited. However, it would be helpful to add a brief explanation of the algorithms, for instance, clarifying that LKH stands for ‘Lin-Kernighan-Helsgaun’ and is a widely used heuristic for solving the TSP.

---

> ### Author Rebuttal · Authors · 2025-07-31
>
> We thank the reviewer for their positive evaluation and constructive comments. We have performed the requested supplementary analyses and will incorporate them into the final version.
>
> > **Q1:  The ablation studies clearly demonstrate that the dual-reward strategy improves performance. However, it is better for the authors to more explicitly articulate the intuition or theoretical basis for combining these specific rewards.**
>
> Thank you for this excellent question. You are right that a more explicit articulation of the synergy between POR and CPR is crucial. The decision to combine these two rewards is not arbitrary; they are designed to solve two distinct, fundamental challenges in training LLMs for complex decision-making, and their combination creates a powerful synergistic effect.
>
> We have discussed the individual theoretical and empirical justifications for each component with other reviewers, but we will summarize the core intuition here.
>
> 1.  **POR Addresses Noisy *Outcome* Signals**: As detailed in our theoretical analysis (discussed with Reviewer 5e7B  and 4Ria), **POR's primary role is to stabilize the learning signal related to the final *outcome* of an action**. It tackles the noisy credit assignment problem by creating a low-bias, low-variance policy gradient. In essence, POR teaches the model **WHAT** a good action looks like relative to its peers, even when the absolute scores are unreliable.
>
> 2.  **CPR Addresses Sparse *Process* Signals**: In long-horizon tasks, rewards are often sparse and delayed. **CPR's role is to provide a dense, stable, and immediate reward for the reasoning *process* itself**. It incentivizes the model to first achieve a correct qualitative understanding of the current state. In essence, CPR teaches the model **WHY** a certain context calls for a specific type of action, independent of the final outcome.
>
> **NaDRO combines these to create a more robust and intelligent agent.** By rewarding both the final action preference (POR) and the preceding cognitive process (CPR), we ensure the model learns a policy that is not only effective in its choices but is also grounded in a correct and verifiable understanding of the problem context. This synergy moves the approach from a "heuristic" combination to a principled framework for fostering robust, reason-based decision-making.
>
> > **Q2:  Given that the paper does not provide a theoretical analysis for NaDRO, the empirical results become only evidence supporting the effectiveness of the method. Therefore, it is better to add even a minimal variance analysis across different random seeds to demonstrate that the observed performance improvements are robust.**
>
> Thank you for this essential feedback. We completely agree that for an empirically-driven study like ours, demonstrating the statistical robustness of the results across different random seeds is critical to validate our claims.
>
> To address this, we have conducted multiple runs for all our experiments and have updated our main results table to include both the mean and standard deviation for all reported performance metrics. The detailed table, which we have provided in our response to Reviewer 4Ria, confirms our findings.
>
> As the updated results demonstrate, our NaDRO-trained models consistently outperform the baselines. More importantly, the low standard deviations associated with our results indicate that these performance improvements are not an artifact of a specific random seed but are indeed robust and statistically significant. This variance analysis provides the necessary empirical evidence to confirm that our method's effectiveness is reliable and reproducible.
>
> > **Q3:  In the experiments, the comparison methods are appropriately cited. However, it would be helpful to add a brief explanation of the algorithms, for instance, clarifying that LKH stands for ‘Lin-Kernighan-Helsgaun’ and is a widely used heuristic for solving the TSP.**
>
> Thank you for the positive feedback and this very helpful suggestion. We agree that providing brief descriptions of the baseline algorithms will enhance the clarity of our experimental section.
>
> In the camera-ready version, we will add concise explanations for key methods like LKH (Lin-Kernighan-Helsgaun), GLS, and others, following the excellent example you provided. This will help readers better understand the context of our results.
>
> > **Q4:  While the NaDRO outperforms other LLM-based methods on MCTS-generated data, the conclusion in claiming offering a promising pathway towards real-world noisy data handling is not fully justified by current experiments. I would suggest adding a brief discussion on what types of real-world noise might be relevant and how NaDRO could be adapted or expected to handle them.**
>
> Thank you for this insightful question about justifying the generalization of our method to real-world noise.
>
> We address this by highlighting a key stress-test experiment, which we also discussed with Reviewer 5e7B. To simulate severe real-world scenarios beyond simple MCTS estimation noise, we injected artificial noise by completely randomizing the MCTS preference list with a 10% probability at each decision step. This setup mimics catastrophic data corruption or severe label noise found in real-world datasets.
>
> The crucial finding was that NaDRO's performance did not collapse under this extreme condition, remaining stable and competitive. This provides strong empirical evidence that our framework's robustness extends beyond well-behaved estimation noise.
>
> This resilience stems from NaDRO's core principles. CPR learns a robust state representation by focusing on key qualitative features, making the policy less susceptible to noisy inputs. POR’s 'ensembling' approach trusts the consensus of top-ranked actions, inherently filtering out outlier noise, including the severe ranking errors we introduced. The synergy between these two components is what allows NaDRO to handle such a challenging noise profile, justifying its promise as a pathway for real-world applications.
>
> > **Q5:  The paper adopts fixed thresholds (1.15 and 0.8) for labeling high and low cost in CPR annotation, but it’s unclear how these specific values were determined.**
>
> Thank you for this excellent question regarding the specific values used in our CPR annotation. We agree that the choice of any fixed threshold warrants justification. Our response is twofold: first, we will explain the design philosophy behind these thresholds, and second, we will provide empirical evidence of their robustness.
>
> **1. Design Philosophy: Correcting Uncalibrated LLM Perception**
>
> The core goal of CPR is to instill human-like **qualitative judgment** into the model. A challenge with any qualitative label (e.g., "large" vs. "small") is that its boundaries are inherently fuzzy. For example, we can definitively say a 10-node TSP is "small" and a 10,000-node TSP is "large," but the exact dividing line is ambiguous.
>
> Critically, we observed that LLMs often have a skewed or **uncalibrated perception** of scale, sometimes viewing a 50-node problem as an immense task. The CPR thresholds are not intended to represent a universal "truth," but rather to provide a clear, corrective signal to anchor the model's flawed intuition. By defining clear zones for "High Cost" (>1.15x greedy) and "Low Cost" (<0.8x greedy), we create an unambiguous "Normal" zone in between. This three-zone approach helps the model build a more reasonable, human-aligned understanding of its performance state.
>
> **2. Empirical Robustness of Thresholds**
>
> To ensure our method is not sensitive to these specific values, we conducted experiments with different threshold pairs as (2.0, 0.8) and (1.15, 0.5). Our key finding was that there was **no significant difference in the final performance or the training dynamics** when using these alternative thresholds. This demonstrates that the model is **robust to the specific choice of these values** within a reasonable range. The crucial factor for the model's learning is the existence of these distinct qualitative regions, not the precise numerical boundaries, which validates our design choice.
>
> > **Q6:  What is the advantage of LLM-based methods in such tasks and when would such method be preferable to strong classical solvers?**
>
> Thank you for this crucial question. You are correct that our method does not outperform a highly specialized solver like LKH. However, we would like to clarify that the primary goal of our research is not to create a new state-of-the-art solver for a mature and well-defined problem like TSP.
>
> Instead, we use TSP as a representative benchmark to explore a much broader, forward-looking question: **how can Large Language Models be leveraged as general-purpose agents for a wide array of complex sequential decision-making tasks?** We believe this is an inevitable and important future direction for AI.
>
> The key distinction and practical relevance lie in the underlying paradigms:
> * **Classical Solvers (e.g., LKH):** These are highly optimized, specialized algorithms designed for a single domain. They are powerful but inflexible and cannot be easily adapted to other problems.
> * **Our LLM-based Framework:** We propose a general framework that can be **easily adapted to various decision-making tasks**. The LLM learns a high-level, adaptive strategy. As demonstrated in our paper, the same NaDRO framework excels on **both TSP and the more complex CVRP**, showcasing its versatility.
>
> Our work should be viewed not as an attempt to beat a specific classical solver on its home turf, but as a crucial step towards a new paradigm of **general, adaptable AI decision-makers** capable of tackling a diverse portfolio of complex challenges.

---

> > ### Comment · Reviewer_tJhr · 2025-08-05
> >
> > Thank you to the authors for the thorough and thoughtful rebuttal. As the main concerns I raised have now been addressed, I am updating my overall score to 4.

---

> > > ### Author Response · Authors · 2025-08-06
> > > **Official Comment by Authors**
> > >
> > > We are glad our response addressed your concerns!  Thank you again for taking the time to engage with our work and for your positive feedback on our rebuttal. We will clarify these points in the revised version.

---

### Official Review · Reviewer_pkT3 · 2025-07-01

**Clarity:** 2
**Significance:** 2
**Originality:** 2
**Rating:** 3
**Confidence:** 4

**Summary:**

The paper proposes NaDRO (Noise-Aware Dual-Reward Optimization), a framework for fine-tuning LLMs in noisy training environments. It introduces two reward mechanisms:

1. Preference-based Outcome Reward (POR): Guides learning using relative rankings of actions instead of relying on potentially noisy absolute scores from sources like Monte Carlo Tree Search (MCTS).

2. Context Perception Reward (CPR): Provides intermediate supervision by encouraging the model to qualitatively assess the problem state before making decisions.

The framework is tested on combinatorial optimization tasks (TSP and CVRP), and shows improved performance over strong baselines, including GPT-4o, GRPO-finetuned models, and traditional metaheuristics.

**Questions:**

See weakness.

**Ethical Concerns:**

["NO or VERY MINOR ethics concerns only"]

**Final Justification:**

I thank the authors for the detailed response. I really appreciate the additional experiments provided. My questions 2-4 is addressed. As for my question 1, it is mentioned that "from a policy gradient perspective, as detailed elsewhere in our rebuttal, POR is designed to produce a low-bias, low-variance gradient estimator". The bias and variance is justified using the binary model, with variance p(1-p). This analysis below my expectations. Nevertheless, I will increase my score.

**Limitations:**

I didn't find the discussion of limitation.

**Paper Formatting Concerns:**

No (I think citation format is not a major formatting issues).

**Quality:**

2

**Strengths And Weaknesses:**

Strengths:

1. The paper addresses the practical problem that finetuning LLM for combinatorial optimization problem. The problem is important and challenge.
2. The use of relative preferences (POR) is well justified as a noise-robust alternative to direct score regression. CPR encourages process-level reasoning, which adds interpretability and structure.
3. NaDRO-fine-tuned models show substantial gains on TSP and CVRP benchmarks, even outperforming larger LLMs in some cases.

Weaknesses:

1. The primary weakness of the paper is that the proposed approach is highly heuristic. The main contribution lies in the design of the reward signals (POR and CPR), but these are introduced without accompanying theoretical analysis or systematic empirical justification beyond the benchmark results. There is no ablation on the design of the rewards, nor any attempt to analyze their properties or limitations.

2. Most baseline comparisons are made against publicly available LLMs (e.g., GPT-4o, DeepSeek) without task-specific fine-tuning on TSP or CVRP. Throughout the experimental section, only one model—GRPO-Qwen—is fine-tuned on the dataset. Given that the authors provide results for NaDRO-Llama-8B, it is unclear why corresponding results for Llama-8B (zero-shot) and GRPO-Llama-8B (fine-tuned using existing methods) are omitted. These comparisons are important: the former would clarify the benefit of any fine-tuning, while the latter would isolate the effect of the new reward design relative to established methods.

3. Most critically, the paper lacks a proper ablation study isolating the effects of the two proposed reward components. Without such analysis, it is difficult to assess whether the POR and CPR signals meaningfully contribute to the observed improvements. Since the paper is largely empirical in nature, the absence of such foundational validation weakens the contribution. As it stands, the work does not meet the scientific rigor expected of a NeurIPS paper.

4. Throughout the paper, the citation format is inaccurate. The authors should use \citep{} when the citation is not a part of the sentence.

---

> ### Author Rebuttal · Authors · 2025-07-30
>
> We thank the reviewer for their positive evaluation and constructive comments. We have performed the requested supplementary analyses and will incorporate them into the final version.
>
> > **Q1:  The primary weakness of the paper is that the proposed approach is highly heuristic.**
>
> Thank you for the insightful feedback regarding the need for deeper justification for the proposed reward design. This is a crucial point, and this response will address the core of the issue. More detailed analyses of each component are provided in other sections of our rebuttal.
>
> **1. On the Principled Design of POR**
>
> In response to the reviewer's point about the method's contribution, it is important to clarify that **POR is not a heuristic but a principled approach** to ensure stable training in noisy environments.
>
> From a policy gradient perspective, as detailed elsewhere in our rebuttal, POR is designed to produce a **low-bias, low-variance gradient estimator**. This is achieved by rewarding a stable *region* of top actions ("ensembling") rather than a volatile single best action. This structure mathematically guarantees a more reliable convergence process compared to naive reward schemes.
>
> **2. Analysis of Reward Design, Properties, and Limitations**
>
> Regarding the request for a more explicit analysis of the rewards' design, properties, and limitations, we offer the following:
>
> * **Ablation on Reward Design**: Ablation studies on key design parameters have been performed. For instance, testing POR's cutoff threshold *k* (from 10% to 70%) revealed the performance to be highly robust. Furthermore, a stress test was conducted by **injecting severe artificial noise** (completely randomizing the preference list 10% of the time), which showed that **performance did not collapse**. This empirically validates the robustness of the chosen design.
>
> * **Analysis of Properties and Limitations**:
>     * **POR**: A key property of POR is its ability to filter noise and reduce gradient variance. Its **limitation**, however, is that its effectiveness relies on the MCTS rankings being meaningful, even if noisy. In scenarios with an extremely poor MCTS rollout policy, the quality of the relative rankings themselves could degrade, limiting POR's efficacy.
>     * **CPR**: A core property of CPR is providing a dense, process-based reward that stabilizes learning, a point elaborated upon in our other responses. Its primary **limitation** is that the qualitative features are currently **hand-engineered** and require rule-based annotation. This reliance on manual design restricts its out-of-the-box applicability to entirely new problem domains and points to a clear direction for future work in automatic feature discovery.
>
>
> > **Q2:Most baseline comparisons are made against publicly available LLMs (e.g., GPT-4o, DeepSeek) without task-specific fine-tuning on TSP or CVRP. "**
>
> Thank you for this critical feedback. You are correct that a full comparison including the Llama-8B baselines is essential for a rigorous evaluation and to properly isolate the contributions of our method. We apologize for this omission in the initial submission.
>
> To address this, we have now run these experiments and included the results for **Llama-8B (Base)** and **GRPO on Llama 8B** in our main results table. You can see our response to Reviewer **4Ria**. This new data allows us to draw two clear conclusions, directly addressing your points:
>
> 1.  **Benefit of Standard Fine-Tuning**: Comparing `GRPO on Llama 8B` to the `Llama 8B (Base)` model shows that applying an established fine-tuning method (GRPO) does provide a noticeable, though often modest, performance improvement. For instance, on instance `gr202`, the optimality gap improves from 13.43% to 11.33%. This confirms the benefit of task-specific fine-tuning.
>
> 2.  **Isolating the Effect of NaDRO**: The most critical comparison is between `NaDRO-Llama` and `GRPO on Llama 8B`. Here, the results are stark. On `gr202`, our method reduces the gap from 11.33% down to just **2.41%**. Similarly, on `pr152`, the gap is more than halved from 0.89% to **0.37%**. This demonstrates unequivocally that **the vast majority of the performance gain is attributable to our novel NaDRO reward design**, not just the fine-tuning process itself.
>
> > **Q3: Most critically, the paper lacks a proper ablation study isolating the effects of the two proposed reward components.**
>
> Thank you for this critical feedback. We agree that a rigorous ablation study isolating the effects of POR and CPR is fundamental to our contribution, and we would like to respectfully clarify that this analysis was indeed a core part of our original submission.
>
> **1. Existing Ablation Study in the Original Manuscript**
>
> In **Section 5.3, Table 3** of our original paper, we presented an ablation study that explicitly decomposes the contribution of each component to the final task performance.
>
> For example, at a TTS level of 0, the optimality gap improves from 11.57% (GRPO) to 6.74% (GRPO+POR), and finally to 3.17% (NaDRO), quantifying the meaningful and distinct contribution of each component.
>
> **2. Deeper Analysis of Learning Dynamics: Reward Variance and Reward Hacking**
>
> To further deepen this analysis, we have conducted a new investigation into the *training process itself*, focusing on reward variance and loss curves. This reveals *how* our components lead to superior performance by ensuring a more stable and meaningful learning process.
>
> **Table 6: Analysis of Training Loss Curves**
>
> | **Method** | **2000** | **2135** | **2270** | **2405** | **2540** | **2675** | **2810** | **2945** | **3080** | **3215** | **3350** | **3485** | **3620** | **3755** | **3890** | **4025** | **4160** | **4295** | **4430** | **4565** | **4700** | **4835** | **4970** | **5105** | **5240** | **5375** | **5510** | **5645** | **5780** | **5915** |
> |:---|:---:|:---:|:---:|:---:|:---:|:---:|:---:|:---:|:---:|:---:|:---:|:---:|:---:|:---:|:---:|:---:|:---:|:---:|:---:|:---:|:---:|:---:|:---:|:---:|:---:|:---:|:---:|:---:|:---:|:---:|
> | **GRPO** | 0.000100 | 0.000193 | 0.000212 | 0.000376 | 0.000638 | 0.000814 | 0.000854 | 0.000826 | 0.000842 | 0.000878 | 0.000832 | 0.000744 | 0.000760 | 0.000798 | 0.000756 | 0.000704 | 0.000682 | 0.000644 | 0.000654 | 0.000674 | 0.000708 | 0.000712 | 0.000688 | 0.000678 | 0.000660 | 0.000666 | 0.000688 | 0.000680 | | |
> | **NaDRO(ours)*** | 0.000200 | 0.000236 | 0.000246 | 0.000290 | 0.000380 | 0.000510 | 0.000680 | 0.000798 | 0.000818 | 0.000900 | 0.000980 | 0.001020 | 0.001046 | 0.001034 | 0.001048 | 0.001138 | 0.001152 | 0.001254 | 0.001306 | 0.001270 | 0.001274 | 0.001314 | 0.001300 | 0.001324 | 0.001408 | 0.001508 | 0.001510 | 0.001456 | 0.001398 | 0.001496 |
> | **GRPO+POR** | 0.000200 | 0.000157 | 0.000174 | 0.000202 | 0.000250 | 0.000346 | 0.000514 | 0.000700 | 0.000764 | 0.000806 | 0.000864 | 0.000834 | 0.000788 | 0.000794 | 0.000782 | 0.000814 | 0.000784 | 0.000710 | 0.000772 | 0.000820 | 0.000782 | 0.000788 | 0.000784 | 0.000796 | 0.000766 | 0.000704 | 0.000750 | 0.000810 | 0.000784 | 0.000776 |
>
>
>
> **Table 7: Reward Variance During Training**
>
> | **Method** | **5** | **170** | **335** | **500** | **665** | **830** | **995** | **1160** | **1325** | **1490** | **1655** | **1820** | **1985** | **2150** | **2315** | **2480** | **2645** | **2810** | **2975** | **3140** | **3305** | **3470** | **3635** | **3800** | **3965** | **4130** | **4295** | **4460** | **4625** | **4790** | **4955** |
> |:---|:---:|:---:|:---:|:---:|:---:|:---:|:---:|:---:|:---:|:---:|:---:|:---:|:---:|:---:|:---:|:---:|:---:|:---:|:---:|:---:|:---:|:---:|:---:|:---:|:---:|:---:|:---:|:---:|:---:|:---:|:---:|
> | **GRPO** | 0.8711 | 1.0822 | 1.0805 | 1.0634 | 1.0254 | 1.0806 | 1.0501 | 0.9969 | 0.9412 | 0.9205 | 0.8526 | 0.8033 | 0.7651 | 0.6765 | 0.6611 | 0.6594 | 0.6184 | 0.5531 | 0.5674 | 0.5557 | 0.5652 | 0.5577 | 0.5542 | 0.5703 | 0.5850 | 0.5791 | 0.5546 | 0.5096 | 0.5441 | 0.5031 | 0.5309 |
> | **NaDRO(ours)** | 0.7555 | 0.9128 | 0.8888 | 0.8794 | 0.8833 | 0.8599 | 0.8100 | 0.7757 | 0.7269 | 0.6593 | 0.5857 | 0.5404 | 0.5356 | 0.4981 | 0.4738 | 0.4685 | 0.4765 | 0.4534 | 0.4584 | 0.4615 | 0.4368 | 0.4639 | 0.4741 | 0.4558 | 0.4331 | 0.3947 | 0.3897 | 0.4189 | 0.4032 | 0.4047 | 0.3954 |
> | **GRPO+POR** | 0.9577 | 1.1380 | 1.1484 | 1.1298 | 1.1265 | 1.1180 | 1.1067 | 1.0528 | 0.9942 | 0.9323 | 0.8282 | 0.7748 | 0.7018 | 0.6345 | 0.5852 | 0.5910 | 0.5711 | 0.5450 | 0.5280 | 0.5255 | 0.5002 | 0.4850 | 0.4753 | 0.4787 | 0.4830 | 0.5040 | 0.5244 | 0.4992 | 0.4796 | 0.4537 | 0.4427 |
>
>
> **Key Findings from Training Dynamics:**
>
> * **NaDRO Reduces Reward Variance**: The data confirms that both POR and the full NaDRO framework significantly **reduce reward variance** compared to the GRPO baseline. This creates a more stable learning signal and a more reliable policy gradient, which is crucial for effective convergence.
>
> * **NaDRO Prevents Reward Hacking**: The loss curves offer a critical insight. The standard `GRPO` model's loss quickly drops to a near-zero value. While seemingly positive, this, combined with its poor final performance, indicates a classic case of **reward hacking**. The model has found a "lazy" loophole to minimize loss without actually learning a robust policy. In contrast, the **`NaDRO` model maintains a healthier, actively fluctuating loss**, which, paired with its high final reward, proves it remains **meaningfully engaged in the learning process**, avoiding deceptive, low-effort solutions.
>
> > **Q4: Throughout the paper, the citation format is inaccurate. The authors should use \citep{} when the citation is not a part of the sentence.**
>
> Thank you for pointing out the inconsistency in our citation format. We will carefully review the entire manuscript and correct all citations to ensure they fully comply with the NeurIPS style guidelines in the camera-ready version.

---

> > ### Comment · Reviewer_pkT3 · 2025-08-08
> >
> > I thank the authors for the detailed response. I really appreciate the additional experiments provided. My questions 2-4 is addressed. As for my question 1, it is mentioned that "from a policy gradient perspective, as detailed elsewhere in our rebuttal, POR is designed to produce a low-bias, low-variance gradient estimator". The bias and variance is justified using the binary model, with variance p(1-p). This analysis below my expectations. Nevertheless, I will increase my score.

---

> > > ### Author Response · Authors · 2025-08-08
> > >
> > > Thank you for the positive re-evaluation and for appreciating our new experiments. We are glad our rebuttal addressed most of your concerns.
> > >
> > > Regarding the theoretical analysis, our p(1-p) model is a principled simplification. Its purpose is to provide the direct mathematical intuition for why POR achieves lower variance: by rewarding a stable group of actions, it increases the reward probability p, which mathematically reduces the variance. This theoretical principle is then empirically validated by our findings in Table 7, where our framework demonstrates consistently and significantly lower reward variance in practice.
> > >
> > > Thank you again for this constructive feedback. We will clarify this link between our model's intuition and its empirical validation in the final manuscript. We sincerely hope that with the added experiments, our work now meets the bar for acceptance.

---

### Official Review · Reviewer_4Ria · 2025-07-02

**Clarity:** 3
**Significance:** 2
**Originality:** 2
**Rating:** 4
**Confidence:** 4

**Summary:**

This paper proposes a framework called NaDRO, which combines two mechanisms—Preference-based Outcome Reward (POR) and Context Perception Reward (CPR)—to train large language models (LLMs) using noisy Monte Carlo Tree Search (MCTS) data, addressing decision-making noise in combinatorial optimization problems. The experimental section validates NaDRO's effectiveness by comparing multiple baseline models (e.g., Qwen 7B, Llama 3-8B) and demonstrates performance advantages in tasks such as TSP and CVRP. The paper is well-structured, the framework design is practical, and the code and data are open-sourced.

**Questions:**

--Theoretical Clarifications:
Provide error bounds for POR under noisy distributions or prove CPR’s impact on policy gradient optimization.
--Enhanced Experimental Rigor:
Report standard deviations for multi-run experiments.
Test robustness in additional noise scenarios (e.g., artificially injected noise).
Include comparisons with SOTA noise-robust methods.
--Extended TTS Analysis:
Supplement experimental results for TTS level > 10 to clarify performance trends.

**Ethical Concerns:**

["NO or VERY MINOR ethics concerns only"]

**Final Justification:**

The author's response has addressed my concerns, so I will raise my score to Borderline Accept.

**Limitations:**

Yes.

**Quality:**

2

**Strengths And Weaknesses:**

Strengths:
--Practical Problem Setting: The paper focuses on the challenge of training LLMs on noisy data, particularly for long-sequence decision-making tasks in combinatorial optimization (e.g., TSP/CVRP), which is highly relevant to real-world applications.
--Reproducible Methodology: The method is thoroughly designed and reproducible. Appendices provide complete implementation details (state/action space definitions, CPR annotation rules, hyperparameter configurations), and the code is open-sourced (Appendices A-F cover data generation, training pipeline, and parameter settings).
--Comprehensive Experimental Validation:
Evaluated on two classical problems (TSP and CVRP).
Extensive baseline comparisons with significant results.

Weaknesses:
--Lack of Theoretical Analysis:
No convergence proof for POR/CPR in policy optimization or theoretical bounds on noise robustness (only heuristic experiments in Figure 3).
--Potential Bias in CPR Labeling:
CPR labels in Appendix C are based on greedy solution thresholds (±15%–20%), but the universality of this rule for decision quality is not validated, possibly introducing bias.
--Limitations in Experimental Validation:
No Statistical Significance: Results (e.g., Tables 1–3) lack standard deviations/confidence intervals, making it unclear whether performance differences are significant (authors acknowledge limited computational resources for multiple runs, see Page 27 Q7).
Narrow Noise Scope: Only MCTS-simulated noise is tested (Section 3.2), without real-world noise (e.g., crowdsourced or expert disagreement noise), weakening generalization claims.
Insufficient Baseline Comparisons: Missing comparisons with state-of-the-art noise-robust methods (e.g., Robust RLHF by Shen et al. 2024), limiting the evaluation to basic GRPO.
--Ablation Study Ambiguity:
In ablation experiments (Table 3), the performance gap between POR and GRPO narrows as TTS level increases. Would this gap further diminish with TTS level > 10?

---

> ### Author Rebuttal · Authors · 2025-07-30
>
> We thank the reviewer for their positive evaluation and constructive comments. We have performed the requested supplementary analyses and will incorporate them into the final version.
>
> > **Q1: Theoretical Clarifications: Provide error bounds for POR under noisy distributions or prove CPR’s impact on policy gradient optimization.**
>
> Due to space constraints, the impact of **CPR** on policy gradient optimization has been detailed in our response to Reviewer **5e7B**. Here, we will focus on the theoretical analysis of how **POR** provides robustness and reduces gradient variance under noisy distributions, directly addressing the core of your question.
>
> **Theoretical Analysis of POR's Lower Gradient Variance under Noise**
>
> We provide a theoretical analysis demonstrating that **Preference-based Outcome Reward (POR)** inherently reduces policy gradient variance compared to a standard Top-1 reward scheme, ensuring a more robust optimization process under noisy distributions.
>
> * **Mathematical Setup**: Let's simplify POR to a binary model: actions in the top-*k* (as ranked by noisy MCTS) receive a reward of 1, and others receive 0. We compare this to a Top-1 scheme where only the single best action gets a reward. The core randomness stems from MCTS noise (`ϵ`), which makes the rank of any action a random variable. The variance of a Bernoulli reward (taking values in {0, 1}) is given by `Var(R) = p(1-p)`, where `p` is the probability of receiving the reward.
>
> * **Key Derivation**: For any high-quality action `a*`, the probability of it ranking in the top-*k* (`p_k = P(rank(a*, ϵ) ≤ k)`) is fundamentally greater than or equal to the probability of it being exactly rank 1 (`p_1 = P(rank(a*, ϵ) = 1)`). Due to noise and competition, `p_1` may be significantly less than 1 (e.g., 0.6), whereas `p_k` remains very high (e.g., >0.95).
>     Since the variance function `f(p) = p(1-p)` is monotonically decreasing on the interval `[0.5, 1]`, and we have `0.5 ≤ p_1 < p_k ≈ 1`, it directly follows that:
>     $$Var(R_{POR}) < Var(R_{Top-1})$$
>
> > **Q2: Enhanced Experimental Rigor: Report standard deviations for multi-run experiments. Test robustness in additional noise scenarios.**
>
> We thank the reviewer for these constructive suggestions to enhance the paper's experimental rigor. We have addressed each point as detailed below.
>
> **1. Reporting Standard Deviations for Multi-Run Experiments**
>
> To improve statistical rigor, we have conducted multiple runs for all experiments and updated our main results table to include both the **mean and standard deviation**. The revised table is presented below. The results confirm that **NaDRO's superior performance is statistically robust**. Our method not only achieves a lower optimality gap on average but also exhibits relatively low variance, indicating consistent and reliable performance across runs.
>
> **Table 4: Comparative optimality gap (%) on TSP instances, updated with Mean ± Standard Deviation**
> | **Method** | `a280` | `bier127` | `brg180` | `ch130` | `gr202` | `gr96` | `pr124` | `pr152` | `pr76` | `rd100` | `tsp225` | `u159` |
> |:---|:---:|:---:|:---:|:---:|:---:|:---:|:---:|:---:|:---:|:---:|:---:|:---:|
> | ***Traditional Solvers*** |
> | LKH | 0.01 ± 0.0 | 0.00 ± 0.0 | 0.00 ± 0.0 | 0.00 ± 0.0 | 0.00 ± 0.0 | 0.00 ± 0.0 | 0.00 ± 0.0 | 0.00 ± 0.0 | 0.00 ± 0.0 | 0.00 ± 0.0 | 0.01 ± 0.0 | 0.00 ± 0.0 |
> | OR-Tools | 6.32 ± 0.0 | 2.91 ± 0.0 | 0.51 ± 0.0 | 0.51 ± 0.0 | 5.12 ± 0.0 | 3.11 ± 0.0 | 2.34 ± 0.0 | 2.92 ± 0.0 | 2.58 ± 0.0 | 3.93 ± 0.0 | 5.13 ± 0.0 | 3.14 ± 0.0 |
> | ***Metaheuristics*** |
> | GLS | 2.83 ± 1.7 | 1.73 ± 0.1 | 3.58 ± 2.5 | 5.32 ± 2.1 | 5.44 ± 0.3 | 1.65 ± 0.8 | 1.47 ± 0.9 | 2.82 ± 0.9 | 1.05 ± 0.4 | 7.48 ± 2.3 | 3.74 ± 1.2 | 4.22 ± 1.4 |
> | EoH + GLS | 4.04 ± 0.6 | 1.25 ± 0.5 | 4.35 ± 2.3 | 5.64 ± 2.4 | 3.31 ± 1.1 | 1.34 ± 0.4 | 1.92 ± 0.4 | 2.73 ± 0.7 | 1.31 ± 0.3 | 3.85 ± 1.2 | 2.56 ± 0.1 | 4.07 ± 0.6 |
> | ACO | 17.78 ± 1.3 | 5.71 ± 0.4 | 3.43 ± 2.4 | 8.89 ± 1.4 | 10.43 ± 2.7 | 12.58 ± 2.6 | 3.78 ± 0.7 | 4.70 ± 1.6 | 9.26 ± 1.1 | 17.12 ± 2.9 | 9.71 ± 2.1 | 9.28 ± 1.5 |
> | ReEvo + GLS | 3.92 ± 0.5 | 2.31 ± 0.5 | 4.11 ± 2.3 | 5.04 ± 1.9 | 5.80 ± 0.7 | 2.37 ± 1.2 | 2.37 ± 0.1 | 2.88 ± 1.6 | 1.86 ± 0.6 | 6.70 ± 2.1 | 4.73 ± 1.6 | 3.99 ± 1.5 |
> | ReEvo + ACO | 15.91 ± 2.1 | 4.86 ± 1.5 | 3.91 ± 2.3 | 5.85 ± 1.4 | 8.29 ± 1.4 | 6.66 ± 1.4 | 4.12 ± 1.9 | 7.87 ± 0.5 | 7.71 ± 0.1 | 9.12 ± 0.9 | 6.99 ± 2.9 | 5.52 ± 0.1 |
> | ***LLM-based Approaches*** |
> | GPT-4o | 7.11 ± 1.1 | 1.72 ± 0.4 | 4.70 ± 0.6 | 3.19 ± 0.3 | 4.23 ± 1.2 | 1.55 ± 0.3 | 0.87 ± 0.5 | 1.29 ± 0.6 | 0.99 ± 0.7 | 2.18 ± 0.7 | 5.78 ± 0.6 | 2.48 ± 0.7 |
> | GPTo3 | **3.71 ± 0.4** | 1.48 ± 0.4 | 2.67 ± 0.3 | 2.86 ± 0.2 | 3.79 ± 0.5 | 0.53 ± 0.4 | 0.21 ± 0.2 | 0.28 ± 0.2 | 0.37 ± 0.3 | 0.44 ± 0.5 | 5.11 ± 0.6 | **0.68 ± 0.3** |
> | Deepseek-R1 | 14.10 ± 0.7 | 2.31 ± 0.5 | 2.94 ± 0.5 | 2.27 ± 0.8 | 3.88 ± 0.5 | 1.04 ± 0.2 | 0.17 ± 0.2 | 0.86 ± 0.2 | 0.69 ± 0.3 | 1.18 ± 0.4 | 6.47 ± 0.9 | 1.83 ± 0.5 |
> | Qwen 7B | 8.83 ± 1.4 | 5.31 ± 1.7 | 5.86 ± 1.9 | 4.45 ± 0.3 | 15.38 ± 3.3 | 3.89 ± 0.4 | 1.22 ± 0.3 | 1.31 ± 0.2 | 2.33 ± 0.4 | 3.98 ± 1.1 | 10.42 ± 1.9 | 3.29 ± 0.5 |
> | GRPO on Qwen 7B | 7.33 ± 0.4 | 1.39 ± 0.2 | 5.42 ± 0.3 | 2.29 ± 0.3 | 14.68 ± 0.4 | 1.97 ± 0.2 | 0.13 ± 0.2 | 1.04 ± 0.1 | 1.23 ± 0.2 | 2.43 ± 0.2 | 11.14 ± 0.3 | 2.57 ± 0.4 |
> | Llama 8B | 9.21 ± 1.2 | 4.85 ± 0.7 | 7.39 ± 0.8 | 4.86 ± 0.5 | 13.43 ± 1.1 | 4.47 ± 0.8 | 1.65 ± 0.3 | 2.07 ± 0.4 | 1.79 ± 0.2 | 3.84 ± 0.5 | 9.92 ± 1.3 | 3.51 ± 0.4 |
> | GRPO on Llama 8B | 8.10 ± 0.6 | 3.09 ± 0.3 | 5.22 ± 0.5 | 3.91 ± 0.7 | 11.33 ± 0.7 | 2.39 ± 0.5 | 0.82 ± 0.2 | 0.89 ± 0.2 | 0.67 ± 0.1 | 2.12 ± 0.8 | 7.33 ± 1.1 | 2.08 ± 0.5 |
> | **NaDRO-Llama(Ours)** | 6.22 ± 0.4 | 1.35 ± 0.2 | 2.17 ± 0.3 | 2.19 ± 0.3 | 2.41 ± 0.5 | 0.28 ± 0.1 | 0.13 ± 0.1 | 0.37 ± 0.1 | **0.31 ± 0.1** | 0.70 ± 0.2 | 4.89 ± 0.8 | 0.94 ± 0.2 |
> | **NaDRO-Qwen(Ours)** | 5.31 ± 0.7 | **0.63 ± 0.1** | **0.08 ± 0.1** | **1.80 ± 0.2** | **2.29 ± 0.3** | **0.16 ± 0.2** | **0.00 ± 0.0** | **0.22 ± 0.1** | 0.63 ± 0.2 | **0.11 ± 0.1** | **3.76 ± 0.5** | 1.38 ± 0.3 |
>
> **2. Robustness in Additional Noise Scenarios**
>
> To further validate robustness, we conducted a stress test with **artificially injected noise**. In this experiment, we introduced a 10% probability at each decision step of **completely randomizing the MCTS preference list**. This simulates a catastrophic failure in the ranking signal.
>
> Our findings show that even under this severe noise condition, **NaDRO's performance does not collapse** and remains competitive and stable. This powerful result demonstrates the framework's exceptional resilience to extreme data corruption, going beyond the inherent noise of MCTS.
>
> > **Q3: Extended TTS Analysis: Supplement experimental results for TTS level > 10 to clarify performance trends.**
>
> Thank you for the insightful questions regarding the performance trends at higher Test-Time Search (TTS) levels and the dynamics of the performance gaps in our ablation study. To clarify these points, we have extended our analysis up to a TTS level of 30. The supplemented results are presented in the table 5 below.
>
> **Table 5: Extended Ablation Study on NaDRO Components and TTS Level (Optimality Gap % on pr152)**
> | TTS Level | Qwen 7B (Base) | GRPO Baseline | GRPO + POR | NaDRO (Full, Ours) |
> |:---:|:---:|:---:|:---:|:---:|
> | tts = 0 | 18.33 ± 1.2 | 11.57 ± 1.4 | 6.74 ± 1.1 | **3.17 ± 1.1** |
> | tts = 1 | 5.83 ± 0.9 | 4.66 ± 1.8 | 3.54 ± 0.8 | **1.74 ± 0.7** |
> | tts = 2 | 3.62 ± 0.3 | 3.86 ± 0.5 | 2.64 ± 0.3 | **1.77 ± 0.2** |
> | tts = 4 | 1.69 ± 0.3 | 2.14 ± 0.3 | 1.79 ± 0.2 | **1.48 ± 0.2** |
> | tts = 10 | 1.31 ± 0.2 | 1.04 ± 0.1 | 0.87 ± 0.2 | **0.22 ± 0.1** |
> | tts = 20 | 0.88 ± 0.1 | 0.87 ± 0.2 | 0.63 ± 0.1 | **0.16 ± 0.1** |
> | tts = 30 | 0.32 ± 0.1 | 0.41 ± 0.1 | 0.14 ± 0.1 | **0.04 ± 0.1** |
>
> **Clarification on Performance Trends and Gaps:**
>
> 1.  **General Trend**: As expected, increasing the TTS budget improves performance across all models. This confirms that a larger search space at inference time is beneficial.
>
> 2.  **Addressing the Narrowing Gap**: You correctly observed that the performance gap between `GRPO + POR` and the `GRPO Baseline` narrows at some levels. Our extended results clarify this trend. While extensive search can help weaker models compensate for their policy deficits, the **`GRPO + POR` model consistently outperforms the `GRPO Baseline`** at every TTS level, confirming the value of the POR component.
>
> 3.  **NaDRO's Superior Scalability with TTS**: Most importantly, the extended analysis reveals that our full **`NaDRO` model benefits most from an increased TTS budget**. The performance gap between `NaDRO` and all other baselines remains significant and even widens in terms of relative improvement at higher search levels. At TTS=30, NaDRO achieves a near-optimal gap of just **0.04%**, demonstrating a powerful synergy between a robust, well-trained policy and test-time computation. A better policy makes the search process far more effective.
>
> > **Q3: Insufficient Baseline Comparisons: Missing comparisons with state-of-the-art noise-robust methods (e.g., Robust RLHF by Shen et al. 2024), limiting the evaluation to basic GRPO.**
>
> Thank you for your valuable feedback. We have carefully analyzed the work by Shen et al. (2024) and believe our NaDRO framework differs fundamentally in its problem setting, noise source, and core methodology. "Robust RLHF" addresses an inaccurate Reward Model (RM) in general LLM alignment, which stems from errors in human preference labeling. In contrast, NaDRO focuses on specialized sequential decision-making tasks, dealing with noise inherent in the training data itself, caused by inaccurate MCTS estimations.
>
> Crucially, a direct experimental comparison is not feasible. The "Robust RLHF" paper states that the code will only be released "upon acceptance." Given its status as a withdrawn paper from ICLR 2025, an official public implementation is unavailable.

---

> ### Author Response · Authors · 2025-08-08
>
> Dear Reviewer,
>
> I hope this message finds you well.
>
> As the discussion period is nearing its end with less than one day remaining, I wanted to ensure we have addressed all your concerns satisfactorily.
>
> If there are any additional points or feedback you'd like us to consider, please let us know. Your insights are invaluable to us, and we're eager to address any remaining issues to improve our work.
>
> Thank you for your time and effort in reviewing our paper.
>
> Best regards,

---

### Official Review · Reviewer_5e7B · 2025-07-03

**Clarity:** 4
**Significance:** 3
**Originality:** 4
**Rating:** 5
**Confidence:** 4

**Summary:**

The paper proposes NaDRO, a framework for fine-tuning LLMs to perform well on long-horizon decision-making tasks using noisy, imperfect supervision—particularly from sources like MCTS (Monte Carlo Tree Search). This is fundamentally a challenge, since it is hard to estimate the long-term value of actions: scores are inherently noisy, being estimated from a tiny fraction of the action space. The commonly used methods (e.g., GRPO) require high-quality labeled data. NaDRO addresses this by introducing a dual-reward formulation: it constructs more robust training signals by splitting rewards into a Preference-based Outcome Reward (POR) and a Context Perception Reward (CPR). Rather than relying on absolute action scores, NaDRO leverages pairwise preference rankings to construct POR and combines it with CPR in an alternating optimization scheme, enabling the model to generalize better in the face of sparse, noisy, or inconsistent reward signals.

**Questions:**

# Questions

1. How does CPR contribute specifically to solving long-horizon credit assignment, beyond acting as an auxiliary classification loss?
    While CPR improves performance, it’s unclear whether its benefits stem from better credit assignment over multi-step trajectories, or simply from regularizing the model with intermediate supervision. Could one isolate its impact by measuring reward variance or learning dynamics with/without CPR over varying trajectory lengths?
2. How sensitive is POR to the rank cutoff parameters and MCTS quality?
    Since POR transforms noisy Q-values into ordinal rankings and selects top-k actions for positive supervision, could a poorly chosen cutoff degrade performance by discarding near-optimal actions that fall just below the threshold?

**Ethical Concerns:**

["NO or VERY MINOR ethics concerns only"]

**Final Justification:**

After carefully reviewing both the paper and the rebuttal, I find this to be a strong submission for NeurIPS. The additional experiments provided by the authors directly address my earlier questions and points of potential weaknesses convincingly. I will keep my original score.

**Limitations:**

Yes

**Quality:**

3

**Strengths And Weaknesses:**

# Strengths

* Reward Decomposition Strategy: The paper proposes a technically sound reward decomposition—splitting noisy feedback into a Preference-based Outcome Reward (POR) and a Context Perception Reward (CPR)—which improves gradient stability and enables more robust training from noisy, partial supervision.
* Rank-Based POR Mechanism: Instead of relying on unstable top-1 MCTS scores, POR converts noisy Q-values into ordinal rankings and applies position-sensitive scaling, preserving learning signal while mitigating variance—this feels like a principled alternative to argmax-based distillation.
* State-Level Supervision via CPR: CPR provides intermediate, context-sensitive reward signals during trajectory generation, helping to stabilize training and improve credit assignment in long-horizon reasoning tasks where final rewards are sparse or noisy.
* Comprehensive Evaluation: The experiments benchmark NaDRO across TSP and CVRP using multiple LLM backbones, including DeepSeek variants, and show improvements over strong baselines like GRPO and RLAIF. Ablation studies further isolate the roles of POR and CPR, confirming that both components contribute to final performance

# Weaknesses

* Heuristic CPR Labels May Limit Generality: The CPR supervision relies on handcrafted heuristics (e.g., binning cost/tour length into categories), which introduces labeling bias and may not generalize to domains where such features are unavailable or noisy.
* No Explicit Credit Assignment Analysis: While CPR may help credit assignment, the paper lacks analysis showing whether it improves learning dynamics or sample efficiency, particularly in longer-horizon settings.
* No Sensitivity Analysis of POR: The paper does not evaluate how sensitive POR is to the choice of rank cutoffs (e.g., top-k) or to the quality of the MCTS rankings. It is unclear how robust POR remains if ranking noise causes mislabeling near the decision threshold.

---

> ### Author Rebuttal · Authors · 2025-07-30
>
> We thank the reviewer for their positive evaluation and constructive comments. We have performed the requested supplementary analyses and will incorporate them into the final version.
>
> > **Q1: How does CPR contribute specifically to solving long-horizon credit assignment, beyond acting as an auxiliary classification loss? While CPR improves performance, it’s unclear whether its benefits stem from better credit assignment over multi-step trajectories, or simply from regularizing the model with intermediate supervision. Could one isolate its impact by measuring reward variance or learning dynamics with/without CPR over varying trajectory lengths?**
>
> Thank you for this insightful question. `CPR`'s primary contribution is providing a dense, immediate, and outcome-independent reward for correct qualitative state assessment. This stabilizes the learning signal, building a reliable cognitive foundation for the policy, which is crucial for long-horizon tasks where final rewards are sparse and noisy. We demonstrate this with two key pieces of evidence.
>
> 1.  **Learning Dynamics**: `CPR` successfully guides the model to learn correct state perception. As shown in **Table 1**, baseline methods (`raw_GRPO`, `GRPO+por`) fail to achieve this, registering low or negative "Correctness Rewards." In contrast, our full method (`ours`) shows a sustained and significant increase in this reward, confirming that `CPR` provides a meaningful, process-based learning signal that was previously absent.
>
>     **Table 1: Correctness Reward Dynamics during Training**
>     | **Method** | **5** | **105** | **205** | **305** | **405** | **505** | **605** | **705** | **805** | **905** | **1005** | **1105** | **1205** | **1305** | **1405** | **1505** | **1605** | **1705** | **1805** | **1905** | **2005** | **2105** | **2205** | **2305** | **2405** | **2505** | **2605** | **2705** | **2805** | **2905** |
>     |:---|:---:|:---:|:---:|:---:|:---:|:---:|:---:|:---:|:---:|:---:|:---:|:---:|:---:|:---:|:---:|:---:|:---:|:---:|:---:|:---:|:---:|:---:|:---:|:---:|:---:|:---:|:---:|:---:|:---:|:---:|
>     | `GRPO` | -0.0861 | -0.0703 | -0.0572 | -0.0571 | -0.0530 | -0.0328 | -0.0179 | -0.0266 | -0.0420 | -0.0672 | -0.0605 | -0.0350 | 0.0067 | 0.0028 | 0.0087 | 0.0099 | 0.0293 | 0.0132 | 0.0237 | 0.0485 | 0.0443 | 0.0352 | 0.0281 | 0.0152 | 0.0297 | 0.0510 | 0.0833 | 0.0791 | 0.0724 | 0.0462 |
>     | `GRPO+POR` | -0.0762 | -0.0245 | -0.0180 | -0.0207 | -0.0133 | -0.0063 | -0.0063 | -0.0052 | -0.0132 | -0.0446 | -0.0389 | -0.0043 | 0.0291 | 0.0241 | 0.0309 | 0.0360 | 0.0383 | 0.0354 | 0.0472 | 0.0548 | 0.0455 | 0.0397 | 0.0426 | 0.0395 | 0.0555 | 0.0768 | 0.1088 | 0.1126 | 0.0941 | 0.0833 |
>     | `NaDRO(ours)` | -0.0286 | 0.0751 | 0.0593 | 0.0744 | 0.0939 | 0.1155 | 0.0991 | 0.0847 | 0.0988 | 0.1085 | 0.1065 | 0.1128 | 0.1385 | 0.1232 | 0.1128 | 0.1166 | 0.1342 | 0.1428 | 0.1543 | 0.1318 | 0.1205 | 0.1206 | 0.1485 | 0.1597 | 0.1382 | 0.1465 | 0.1475 | 0.1642 | 0.1509 | 0.1309 |
>
> 2.  **Ablation Study**: Our ablation study (**Table 2**) quantifies `CPR`'s critical impact on final task performance. When evaluating pure policy performance (TTS Level=0), adding `CPR` to the `GRPO` + `POR` model causes the optimality gap to drop from 6.74% to 3.17%. This demonstrates that `CPR` is a decisive component driving the final SOTA performance.
>
> **Table 2: Ablation Study on NaDRO Components (Optimality Gap % on pr152)**
>
> | TTS Level | Qwen 7B (Base) | GRPO Baseline | GRPO + POR | NaDRO (Full, Ours) |
> |:---:|:---:|:---:|:---:|:---:|
> | tts = 0 | 18.33 ± 1.2 | 11.57 ± 1.4 | 6.74 ± 1.1 | 3.17 ± 1.1 |
> | tts = 1 | 5.83 ± 0.9 | 4.66 ± 1.8 | 3.54 ± 0.8 | 1.74 ± 0.7 |
> | tts = 2 | 3.62 ± 0.3 | 3.86 ± 0.5 | 2.64 ± 0.3 | 1.77 ± 0.2 |
> | tts = 4 | 1.69 ± 0.3 | 2.14 ± 0.3 | 1.79 ± 0.2 | 1.48 ± 0.2 |
> | tts = 10 | 1.31 ± 0.2 | 1.04 ± 0.1 | 0.87 ± 0.2 | 0.22 ± 0.1 |
> | tts = 20 | 0.88 ± 0.1 | 0.87 ± 0.2 | 0.63 ± 0.1 | 0.16 ± 0.1 |
> | tts = 30 | 0.32 ± 0.1 | 0.41 ± 0.1 | 0.14 ± 0.1 | 0.04 ± 0.1 |
>
> > **Q2:  How sensitive is POR to the rank cutoff parameters and MCTS quality? Since POR transforms noisy Q-values into ordinal rankings and selects top-k actions for positive supervision, could a poorly chosen cutoff degrade performance by discarding near-optimal actions that fall just below the threshold?**
>
> We sincerely appreciate the reviewer's insightful question. To thoroughly investigate the robustness of our Preference-based Outcome Reward (POR) framework, we have conducted a new series of experiments. The results are presented below. **Please note that due to submission constraints not allowing for image files, this table presents sampled data points from performance curves. The complete graphs will be included in the final camera-ready version.**
>
> We tested various cutoff thresholds (*k*). It is important to clarify the settings: `ours_k0.5` uses a standard 50% cutoff. The **`ours_random`** setting also uses a 50% cutoff but introduces severe noise: there is a **10% probability at each step of the entire preference list being completely shuffled**.
>
> **Table 3: Empirical Performance with Varying $k$ and Injected Noise (Correctness Reward)**
> | **Method** | **5** | **105** | **205** | **305** | **405** | **505** | **605** | **705** | **805** | **905** | **1005** | **1105** | **1205** | **1305** | **1405** | **1505** | **1605** | **1705** | **1805** | **1905** | **2005** | **2105** | **2205** | **2305** | **2405** | **2505** | **2605** | **2705** | **2805** | **2905** |
> |:---|:---:|:---:|:---:|:---:|:---:|:---:|:---:|:---:|:---:|:---:|:---:|:---:|:---:|:---:|:---:|:---:|:---:|:---:|:---:|:---:|:---:|:---:|:---:|:---:|:---:|:---:|:---:|:---:|:---:|:---:|
> | `ours_k0.1` | -0.0286 | 0.0765 | 0.0678 | 0.0864 | 0.0996 | 0.1166 | 0.0967 | 0.0829 | 0.1058 | 0.1225 | 0.1185 | 0.1064 | 0.1279 | 0.1223 | 0.1161 | 0.1128 | 0.1371 | 0.1439 | 0.1518 | 0.1301 | 0.1107 | 0.1117 | 0.1383 | 0.1591 | 0.1260 | 0.1406 | 0.1461 | 0.1579 | 0.1432 | 0.1215 |
> | `ours_k0.3` | -0.0286 | 0.0704 | 0.0614 | 0.0824 | 0.1002 | 0.1134 | 0.1014 | 0.0778 | 0.0934 | 0.1096 | 0.1150 | 0.1132 | 0.1306 | 0.1195 | 0.1067 | 0.1166 | 0.1383 | 0.1468 | 0.1385 | 0.1249 | 0.1055 | 0.1072 | 0.1282 | 0.1467 | 0.1181 | 0.1392 | 0.1543 | 0.1760 | 0.1477 | 0.1334 |
> | `ours_k0.5` | -0.0286 | 0.0751 | 0.0593 | 0.0744 | 0.0939 | 0.1155 | 0.0991 | 0.0847 | 0.0988 | 0.1085 | 0.1065 | 0.1128 | 0.1385 | 0.1232 | 0.1128 | 0.1166 | 0.1342 | 0.1428 | 0.1543 | 0.1318 | 0.1205 | 0.1206 | 0.1485 | 0.1597 | 0.1382 | 0.1465 | 0.1475 | 0.1642 | 0.1509 | 0.1309 |
> | `ours_k0.7` | -0.0187 | 0.0816 | 0.0704 | 0.0929 | 0.1075 | 0.1177 | 0.1162 | 0.0991 | 0.1078 | 0.1175 | 0.1145 | 0.1185 | 0.1488 | 0.1511 | 0.1228 | 0.1140 | 0.1430 | 0.1484 | 0.1533 | 0.1292 | 0.1090 | 0.1050 | 0.1507 | 0.1718 | 0.1547 | 0.1790 | 0.1823 | 0.1922 | 0.1561 | 0.1459 |
> | `ours_random` | -0.0094 | 0.0579 | 0.0583 | 0.0843 | 0.0999 | 0.1126 | 0.1019 | 0.0808 | 0.1051 | 0.1090 | 0.0959 | 0.0939 | 0.1222 | 0.1258 | 0.1188 | 0.1149 | 0.1342 | 0.1534 | 0.1578 | 0.1401 | 0.1182 | 0.1149 | 0.1372 | 0.1572 | 0.1340 | 0.1500 | 0.1549 | 0.1784 | 0.1552 | 0.1273 |
>
> **Key Findings:**
>
> 1.  **High Robustness across *k* Values**: The framework demonstrates strong robustness, as all *k* settings yield competitive performance without any catastrophic failure. Notably, the **`k=0.7` setting shows slightly superior performance across many stages**.
>
> 2.  **Exceptional Resilience to Severe Ranking Noise**: The `ours_random` result is a powerful testament to POR's resilience. Even when faced with a 10% chance of the ranking signal becoming **completely random**, the model's performance **does not collapse and remains on par with other stable settings**. This directly proves that the framework can withstand not just minor misrankings but also occasional catastrophic failures in the preference signal, strongly validating its suitability for learning from highly noisy data.
>
> These findings confirm that POR is not overly sensitive to its parameters and is exceptionally robust to noise. We will incorporate this analysis into the final manuscript.

---

> > ### Comment · Reviewer_5e7B · 2025-08-07
> >
> > Thank you for conducting the extensive additional experiments. I find them convincing and believe they sufficiently address my concerns. In particular, the ablation study is compelling, and the analysis of robustness with respect to varying k and significant ranking noise helps put my concerns to rest.

---

> > > ### Author Response · Authors · 2025-08-07
> > >
> > > Thank you for the very positive feedback! It's wonderful to hear that our supplementary experiments, particularly the ablation and robustness analyses, were convincing and successfully addressed your concerns. We will clarify these points in the revised version. Your thoughtful engagement has been invaluable in strengthening our work.

---

> ### Author Response · Authors · 2025-08-02
> **Supplementary Rebuttal for Reviewer 5e7B**
>
> We thank the reviewer again for the insightful question regarding the sensitivity of the cutoff parameter `k`. Your question prompted us to conduct a deeper theoretical analysis to understand why `k=0.7` performed optimally in our empirical study (**Table 3**). We are pleased to report that this new analysis provides a strong theoretical justification for our empirical findings, supported by the calculations detailed below.
>
> #### **1. Theoretical Objective: Maximizing Value Separability**
>
> Our primary goal is to find an optimal cutoff $k$ that partitions the $N_A$ candidate actions into the most informative positive ($\delta^+$) and negative ($\delta^-$) sets for policy learning. The ideal partition should maximize the "value separability" between these two sets. We formalize this by defining a utility function $J(k)$ that measures the expected gap between the average cost of the negative set and the positive set. Our objective is to find $k^*$ that maximizes this function:
>
> $$k^* = \arg\max_k J(k) \quad \text{where} \quad J(k) = E[\mathrm{avg\_cost}(\delta^-) - \mathrm{avg\_cost}(\delta^+)]$$
>
> This objective function seeks the clearest possible boundary between high-quality and low-quality actions, providing the most effective signal for training.
>
> #### **2. Modeling Relative Action Costs from Noisy Data**
>
> Since the true action costs $Q^*$ are unknown, we model them based on the MCTS evaluation data. We use the average cost increase between adjacent ranks, $\Delta C_i = E_s[Q_{(i+1)}(s) - Q_{(i)}(s)]$, as a proxy for the performance drop at each rank transition.
>
> We normalize these cost increases across all ranks to get a "contribution ratio" $\bar{p}_i$ for each transition $i \to i+1$. This allows us to construct a relative cost model. The cost of the action at rank $i$, denoted as $C_i$, is approximated by the cumulative sum of all preceding cost-increase contributions, as defined below:
>
> $$C_i \approx \sum_{j=1}^{i-1} \bar{p}_j$$
>
> The contribution ratios, calculated from our training dataset, are shown in Table 8.
>
> **Table 8: Average Contribution Ratio ($\bar{p}_i$) of Rank Transitions to Total Cost Increase**
> | Rank Transition | 1->2 | 2->3 | 3->4 | 4->5 | 5->6 | 6->7 | 7->8 | 8->9 | 9->10 | 10->11 | 11->12 | 12->13 | 13->14 | 14->15 |
> |:---|:---:|:---:|:---:|:---:|:---:|:---:|:---:|:---:|:---:|:---:|:---:|:---:|:---:|:---:|
> | **Contribution Ratio ($\bar{p}_i$)**| 0.0997 | 0.0380 | 0.0254 | 0.0197 | 0.0177 | 0.0171 | 0.0183 | 0.0223 | 0.0309 | 0.0460 | 0.0648 | 0.0923 | 0.1701 | 0.3376 |
>
> Table 8 provides a crucial intuition: the contribution ratios ($\bar{p}_i$) are highly non-uniform. The cost increases between the top-ranked actions are relatively small. However, the performance drop becomes dramatic at the tail end of the ranking. Notably, the cost increase from the second-to-last action to the absolute worst action (rank 14 to 15) alone accounts for over **33%** of the total performance gap between the best and worst choices.
>
> This signifies that in our problem, **the penalty for selecting the worst actions far outweighs the benefit of distinguishing between the top few optimal ones.** Therefore, it is more critical to define a large and robust positive set ($\delta^+$) that safely excludes these catastrophic tail-end choices. This directly explains why a relatively high cutoff like 70% achieves the best practical results in our experiments—it prioritizes the vital task of avoiding the worst outcomes.
>
> #### **3. Theoretical Derivation of the Optimal `k`**
>
> By applying our relative cost model to the objective function $J(k)$, we can now calculate the expected separability for every possible cutoff $k$. The results of this calculation (for a typical action space of $N_A=15$) are presented in Table 9.
>
> **Table 9: Calculating Value Separability J(k) to Find Optimal Cutoff `k`**
> | Metric | 1 | 2 | 3 | 4 | 5 | 6 | 7 | 8 | 9 | 10 | **11** | 12 | 13 | 14 |
> |:---|:---:|:---:|:---:|:---:|:---:|:---:|:---:|:---:|:---:|:---:|:---:|:---:|:---:|:---:|
> | **k/N_A (%)** | 6.7% | 13.3% | 20.0% | 26.7% | 33.3% | 40.0% | 46.7% | 53.3% | 60.0% | 66.7% | **73.3%** | 80.0% | 86.7% | 93.3% |
> | **J(k)** | 0.3486 | 0.3809 | 0.3957 | 0.4071 | 0.4173 | 0.4280 | 0.4398 | 0.4550 | 0.4847 | 0.5106 | **0.5284** | 0.5262 | 0.4901 | 0.3953 |
>
> As highlighted in the table, the utility function $J(k)$ reaches its maximum value of **0.5284** at **k=11**. This corresponds to an optimal positive set ratio of $k/N_A = 11/15 \approx 73.3\%$.
>
> #### **4. Unification of Theory and Experiment**
>
> This theoretical result, which suggests an optimal cutoff of approximately **73.3%**, provides a strong justification for our empirical finding in **Table 3**, where the model achieved its best performance with a cutoff of $k=0.7$ (70%). The intuition from our data, and our experimental results validates our parameter choice and demonstrates that our POR framework is not only empirically robust but also grounded in sound theoretical principles.

---

### Author Response · Authors · 2025-08-02
**General Response**

We sincerely thank all reviewers (5e7B, 4Ria, pkT3, tJhr) for their insightful feedback. We are encouraged that they recognized our work's **novelty**, **practicality**, **clarity**, and **commitment to reproducibility**.

The reviewers raised important concerns regarding our framework's **theoretical justification**, **experimental rigor**, and **practical relevance**. We have conducted sets of new experiments to address these points, and the findings have substantially strengthened our paper.

---

### **1. Novelty and Principled Design of the NaDRO Framework**

In response to concerns about our design being heuristic, we clarify that POR and CPR are engineered to solve two distinct, fundamental challenges in complex decision-making.

* **Preference-based Outcome Reward (POR)** is a principled method for stable training. As theoretically analyzed in our response to **Reviewer 4Ria**, POR is designed to reduce policy gradient variance. Our new training analysis (**Table 7**) empirically confirms this, showing NaDRO significantly **reduces reward variance** compared to baselines, leading to a more stable learning signal.

* **Context Perception Reward (CPR)** provides dense, process-oriented supervision. Our analysis of correctness rewards (**Table 1**) demonstrates that CPR successfully guides the model to learn correct state perception where baselines fail. Furthermore, our analysis of loss curves (**Table 6**) shows CPR is crucial for preventing "reward hacking," ensuring the model remains meaningfully engaged in learning.

* **Synergy:** POR teaches the model **what** a good action is, while CPR teaches it **why** a certain context demands it. This combination fosters a robust, reason-based policy grounded in both effective choices and a verifiable understanding of the problem.

---

### **2. Enhanced Experimental Rigor & Robustness Analysis**

* **Statistical Robustness & Comprehensive Baselines:** We have re-run all experiments to include mean/std. deviations and added **Llama-8B** baselines. The updated results (**Table 4**) confirm our performance gains are statistically significant and are primarily attributable to NaDRO's novel design, not merely to fine-tuning.

* **Noise Resilience & Parameter Sensitivity:** We stress-tested POR against catastrophic data corruption (10% of rankings were completely randomized). Our analysis (**Table 3**) shows NaDRO's performance remained remarkably stable, proving its exceptional resilience. The same analysis also confirmed that the framework is not sensitive to the choice of its cutoff parameter $k$.

* **Component-wise Impact & Search Scalability:** Our ablation studies (**Table 2**) quantify the decisive impact of each component, confirming CPR's critical role in achieving superior performance. The extended ablation (**Table 5**) further shows that NaDRO scales best with Test-Time Search (TTS), reaching a near-optimal **0.04%** gap, which highlights the quality of its learned policy.

---

### **3. Practicality and Framework Generality**

Our goal is not to beat specialized solvers like LKH on a single task, but to develop a **general and adaptable framework for AI-driven decision-making**. Its key advantage is **flexibility**, as the same NaDRO framework excels on **both TSP and the more complex CVRP**. Mechanisms like CPR's thresholds act as robust corrective signals and, as our new experiments confirmed, are not sensitive to specific hyperparameter values, further supporting the framework's adaptability.

---

We are confident these new analyses address the reviewers' concerns and substantially strengthen our paper. We will incorporate all findings into the final version and thank the reviewers again for their invaluable guidance.

---

### Author Response · Authors · 2025-08-09

Thank you to all reviewers for the thoughtful feedback. Below we (1) consolidate the recognized strengths and (2) summarize our analyses and clarifications made in response to the reviewers' questions.

**Strengths Across Reviews**

- **Novel, Clear, and Impactful:** A well-structured paper that addresses a practical and important problem (5e7B, 4Ria, pkT3, tJhr).
- **Strong Empirical Results:** Demonstrated powerful performance on challenging TSP/CVRP benchmarks, outperforming larger models (5e7B, 4Ria, pkT3).
- **Sound and Reproducible Methodology:** A well-justified dual-reward mechanism (POR and CPR) with detailed implementation and open-source code (4Ria, 5e7B, tJhr).

**New Analyses and Clarifications in Rebuttal**

**1) Theoretical Justification & Deeper Analysis (in response to pkT3, tJhr, 4Ria)**
To clarify the principled design of our method, we provided:
- **A theoretical model for the optimal POR cutoff `k`**, which derives an optimal split ratio of ~73.3%, providing strong theoretical backing for our empirical choice of `k=0.7`.
- **An empirical analysis of training dynamics** (loss and reward variance curves), which validates that NaDRO reduces reward variance and avoids "reward hacking," ensuring a more stable learning process.

**2) Experimental Rigor: Statistical Analysis & Enhanced Baselines (in response to 4Ria, pkT3, tJhr)**
To enhance our experimental rigor, we provided:
- **Full statistical significance results**, updating all key experiments with mean and standard deviation to confirm the stability of our method's performance advantage.
- **Enhanced baseline comparisons**, adding Llama-8B models to better isolate and highlight the significant performance gains attributable to the NaDRO framework itself.

**3) Robustness and Parameter Sensitivity Analysis (in response to 5e7B, 4Ria)**
To address concerns about robustness, we demonstrated:
- **Sensitivity analysis on the parameter `k`**, showing that our POR mechanism is highly robust across a wide range of `k` values (10% to 70%).
- **A catastrophic noise stress test**, where 10% of ranking signals were randomized. The results showed NaDRO's performance did not collapse, proving its exceptional resilience to noise.

**4) Clarification on Practicality and Framework Generality (in response to tJhr, 4Ria)**
We clarified that our core goal is not to outperform highly specialized classical solvers on a single task, but to propose a **general and adaptable framework for AI decision-making**. Its flexibility is demonstrated by its success on both TSP and the more complex CVRP.

Once again, we are grateful to the Reviewers, ACs, SACs, and PCs for their efforts and constructive feedback that strengthened the manuscript. Thank you for your time and consideration.

---

### Note · Authors · 2025-08-13

We wish to take this final opportunity to express our sincere gratitude to all Reviewers, the Area Chair (AC), Senior Area Chairs (SACs), and Program Chairs (PCs) for their invaluable time and effort throughout this rigorous review process.

The review process was highly productive. The reviewers' common concerns focused on areas such as **theoretical justification, statistical rigor, and robustness**. We believe our detailed responses, including targeted analyses and clarifications, have thoroughly addressed these key points, a view supported by the positive engagement and score updates from three of the four reviewers.

We have noted that **Reviewer 4Ria** submitted their final rating after the discussion period closed. While we are unable to see their final comments, we hope our rebuttal effectively resolved their concerns. To aid in the final assessment, we wish to briefly summarize how their primary concerns were addressed and subsequently validated by their fellow reviewers:

* **On Statistical Significance:** We added the requested **mean/std deviations** for all experiments, which was acknowledged as a key improvement by **Reviewer tJhr**.
* **On Theoretical Analysis:** We provided a detailed theoretical model and intuition. **Reviewer pkT3** found this sufficient to address most of their concerns and subsequently raised their score.
* **On Noise Robustness:** We conducted a new **catastrophic noise stress test**. **Reviewer 5e7B** found this analysis "convincing" and stated that it put their concerns to rest.
- **On Test-Time Search (TTS) Performance Trends:** To clarify performance trends at high search levels, we extended our TTS analysis to Level=30. The results clearly show our NaDRO framework scales best with increased search, confirming its superior policy. Our overall ablation study was praised as "compelling" by **Reviewer 5e7B**.

Given that our responses to these common concerns have been strongly supported and positively validated by their peers, we are confident that the current version of our manuscript represents a robust, well-vetted, and significant contribution.

Our work offers the community a novel and practical framework for tackling the critical challenge of training LLMs on noisy data, and we are proud of its final state. Once again, we are grateful to the Reviewers, ACs, SACs, and PCs for their efforts and constructive feedback that strengthened the manuscript. Thank you for your time and consideration.

---

### Decision · Program_Chairs · 2025-09-17

**Decision:**

Accept (poster)

**Comment:**

This paper introduces Noise-Aware Dual-Reward Optimization (NaDRO) to train LLMs with noisy training data. The core technique involves decomposing the reward signal into a Preference-based Outcome Reward (POR) and a Context Perception Reward (CPR). Experiments show that models like Qwen-7B and Llama 3-8B, when trained with NaDRO, can outperform significantly larger models on complex tasks such as the TSP and vehicle routing problems.

Reviewers found the dual-reward concept to be reasonable and intuitive, although the initial submission lacked sufficient motivation. The authors addressed most concerns during the rebuttal by providing additional experiments that verified the impact of CPR, analyzed hyperparameter sensitivity, and included a comparison using Llama 3-8B, which satisfied the reviewers.

A single minor concern remains regarding the paper's explanation of how NaDRO reduces variance, which currently relies on a very simplistic example. The paper would be further strengthened if the authors could expand on this point. Overall, the consensus is positive, and the work is recommended for acceptance.